# Seasonal variations of *Q. pubescens* isoprene emissions from an *in natura* forest under drought stress and sensitivity to future climate change in the Mediterranean area.

Anne-Cyrielle Genard-Zielinski[1,2], Christophe Boissard[2,3], Elena Ormeño[1], Juliette Lathière[2], Ilja M. Reiter[4], Henri Wortham[5], Jean-Philippe Orts[1], Brice Temime-Roussel[5], Bertrand Guenet[2], Svenja Bartsch[2], Thierry Gauquelin[1] and Catherine Fernandez[1]

[1]Institut Méditerranéen de Biodiversité et d'Ecologie marine et continental, Aix Marseille Université-CNRS-IRD-Univ. Avignon, Marseille, 13331CEDEX 3, France

[2]Laboratoire des Sciences du Climat et de l'Environnement, LSCE/IPSL, CEA-CNRS-UVSQ, Université Paris-Saclay, Gif-sur-Yvette, 91191, France

[3]Université Paris Diderot Paris 7, Paris, 75013, France

[4]Fédération de Recherche 'Ecosystèmes Continentaux et Risques Environnementaux', CNRS FR 3098 ECCOREV, Technopôle de l'environnement Arbois-Méditerranée, Aix-en-Provence, 13545, France

[5]Laboratoire de Chimie de l'Environnement, Aix Marseille Université, CNRS, UMR 7376, Marseille, 13331, France

*Correspondence to* : Christophe Boissard, christophe.boissard@lsce.ipsl.fr

**Abstract.** At a local level, biogenic isoprene emissions can greatly affect the air quality of urban areas surrounded by large vegetation sources, such as in the Mediterranean region. The impacts of future warmer and drier conditions on isoprene emissions from Mediterranean emitters are still under debate. Seasonal variations of *Q. pubescens* gas exchange and isoprene emission rates (ER) were studied from June 2012 to June 2013 at the O₃HP site (French Mediterranean) under natural (ND) and amplified (AD, 32%) drought. While AD significantly reduced stomatal conductance to water vapour throughout the research period excepting August, it did not significantly preclude $CO_2$ net assimilation, which was lowest in summer ($\approx$ -1 $\mu mol_{CO2}$ m$^{-2}$ s$^{-1}$). ER followed a significant seasonal pattern regardless of drought intensity, with mean ER maxima of 78.5 and 104.8 $\mu gC$ $g_{DM}^{-1}$ h$^{-1}$ in July (ND) and August (AD) respectively, and minima of 6 and <2 $\mu gC$ $g_{DM}^{-1}$ h$^{-1}$ in October and April respectively. Isoprene emission factor increased significantly by a factor of 2 in August and September under AD (137.8 and 74.3 $\mu gC$ $g_{DM}^{-1}$ h$^{-1}$) compared with ND (75.3 and 40.21 $\mu gC$ $g_{DM}^{-1}$ h$^{-1}$), but no significant changes occurred on ER. Aside from the June 2012 & 2013 measurements, MEGAN2.1 model was able to assess the observed ER variability only when its soil moisture activity factor $\gamma_{SM}$ was not operating, and regardless of the drought intensity; in this case more than 80% and 50% of ER seasonal variability was assessed in the ND and AD respectively. We suggest that a specific formulation of $\gamma_{SM}$ be developed for drought adapted isoprene emitter, according with that obtained for *Q. pubescens* in this study ($\gamma_{SM}$ =

$0.192e^{51.93\ SW}$, with SW the soil water content). An isoprene algorithm (*G*14) was developed using an optimised artificial neural network trained on our experimental dataset (ER + O$_3$HP climatic and edaphic parameters cumulated over 0 to 21 days prior to the measurements). *G*14 assessed more than 80% of the observed ER seasonal variations, whatever the drought intensity. ER$_{G14}$ was more sensitive to higher (0 to -7 days) frequency environmental changes under AD in comparison to ND. Using IPCC RCP2.6 and RCP8.5 climate scenarios, and SW and temperature as calculated by the ORCHIDEE land surface model, ER$_{G14}$ was found to be mostly sensitive to future temperature, and nearly not to precipitation decrease (an annual increase of up to 240% and at the most 10% respectively in the most severe scenario). The main impact of future drier conditions in the Mediterranean was found to be an enhancement (+40%) of isoprene emissions sensitivity to thermal stress.

## 1    Introduction

A large number of Mediterranean deciduous and evergreen trees produce and release isoprene (2-methyl-1,3-butadiene, C$_5$H$_8$). Under non-stress conditions, only 1-2% of the carbon recently assimilated is emitted as isoprene, whereas under stress conditions such as water scarcity this value can reach up to 20-30% (*Quercus pubescens*, Genard-Zielinski et al., 2014). Although the role of isoprene remains a subject of debate, it seems likely that C$_5$H$_8$ helps plants to optimise CO$_2$ assimilation during temporary and mild stresses, especially during the growing and warmer periods (Brili et al., 2007; Loreto and Fineschi, 2015). The major role of isoprene in plant defence probably explains its large annual global emissions (440-660 TgC.y$^{-1}$, Guenther et al., 2006), forming the largest quantity of all Biogenic Volatile Organic Compounds (BVOC) emitted. Although present in the atmosphere at the ppb or ppt level, isoprene has a broad impact on atmospheric chemistry, both in the gas phase (especially in the O$_3$ budget of some urbanised areas, Atkinson and Arey, 2003) and in the particulate-phase (secondary organic aerosols formation, Goldstein and Steiner, 2007), and hence on biosphere-atmosphere feedbacks. For instance, in the Mediterranean area, Curci et al. (2009) showed that isoprene could be responsible for the production of 4 to 6 ppbv of ozone between June and August, representing 16-20% of total ozone. Given the broad impacts of isoprene on atmospheric chemistry, considerable efforts have been made to (i) understand the physiological mechanisms responsible for isoprene synthesis and emission and the different environmental parameters that control their variability, in order to (ii) develop isoprene emission models that can account for the broadest possible range of environmental conditions.

Thus, it has extensively been shown that under non-stressful conditions, isoprene synthesis and emission are closely connected and primarily depend upon light and temperature conditions (Guenther et al., 1991, 1993). In contrast, under environmental stress, isoprene emission and synthesis are uncoupled in a way that is not fully understood and hence still under debate (Affek and Yakir, 2003; Peñuelas and Staudt, 2010). Indeed, although some authors have identified an increase in isoprene emission under mild water stress (Sharkey and Loreto, 1993; Funk et al., 2004; Pegoraro et al., 2004; Genard-Zielinski et al., 2014), others have reported the opposite (Bruggemann and Schnitzler, 2002; Rodriguez-Calcerrada et al., 2013; Tani et al., 2011).

Concerning the modelling of isoprene emission variations, two main approaches have been considered so far: (i) empirically based parameterisations to represent observed emission variations in relation to easily accessible environmental drivers, and (ii) process-based relationships built on the understanding of the ongoing biological regulation (see Ashworth et al., 2013). Both types of model are adapted for global/regional modelling, but the former are more commonly used for atmospheric applications, especially for air quality exercises for which mechanistic models remain far too complex. Indeed, whilst Grote et al. (2014) have indicated that such models are fairly effective in accounting for the mild stress effects on seasonal isoprene variations of *Quercus ilex*, the large number of necessary describing parameters continues to represent an obstacle for their broad and routine use in air quality (Ashworth et al., 2013). Moreover, the development of BVOC empirical emission models, and especially of the most widely used empirical model, MEGAN (Model of Emissions of Gases and Aerosols from Nature, Guenther et al., 2006, 2012), was partly based on measurements carried out under 'optimum' growing conditions and/or obtained from very few emitters. Therefore, if they depict a fair picture of the general level and global distribution of BVOC emission, they remain somewhat deficient in accounting for a large range of stress conditions. When used for air quality monitoring applications, such a bias intrinsic to the model can significantly weaken air quality forecasts in areas that are greatly influenced by biogenic sources (von Kuhlmann et al., 2004; Chaxel and Chollet, 2009). Concerning the impact of drought strees, the inclusion of the soil moisture effect on isoprene emission in MEGAN was derived from a sole drought study made on *Populus deltoides* (Pegoraro et al., 2004). Validation regarding a broader range of environmental conditions (including stress conditions) and emitters is necessary. Weaknesses in accounting for the impact of drought can be detrimental to isoprene emission inventories, especially when undertaken in areas that are covered with a large quantity of high isoprene emitters and that

are subject to frequent drought episodes, like the Mediterranean region. Moreover, in addition to a predicted temperature increase of between 1.5 and 3°C, climate models over this area predict an amplification of the natural drought during summers due to a reduction in precipitation that could locally reach up to 30% by the year 2100 (Giorgi and Lionello, 2008; Intergovernmental Panel on Climate Change, 2013; Polade et al., 2014). Owing to the close interactions between air pollution over large Mediterranean urban areas and strong BVOC emissions from nearby vegetation, the potential impacts of future climatic changes on isoprene emissions represents an acute environmental issue needing to be addressed (Chameides et al., 1988; Atkinson and Arey, 1998; Calfapietra et al., 2009; Pacifico et al., 2009). Within this context, a recent study has underlined the importance of monitoring over a long period both isoprene emissions and soil moisture in water limited ecosystems (Zheng et al., 2015). Since *Q. pubescens* Willd. is the second largest isoprene emitter in Europe (and foremost in the Mediterranean zone) (Keenan et al., 2009), it represents an ideal model species by which to investigate isoprene emission variability under drought conditions.

The objectives of this study were (i) to investigate *in natura* the influence of natural (ND) and amplified (AD) drought on *Q. Pubescens* seasonal gas exchanges ($CO_2$, $H_2O$), and in particular, isoprene emission rates (ER); (ii) to test and compare two empirical emission models, MEGAN2.1 (Guenther et al., 2012) and *G*14 (this study), in assessing seasonal ER variability under different drought intensities, and (iii) to evaluate the sensitivity of ER to future climatic changes (warming and precipitation reduction) based on two extreme IPCC scenarios: RCP2.6 (moderate) and RCP8.5 (extreme).

## 2   Materials and methods

### 2.1   Experimental site O₃HP

Experimental data were obtained at the O₃HP site (Oak Observatory at the Observatoire de Haute Provence, 5°42'44" E, 43°55'54" N). This site constitutes part of the French national network SOERE F-ORE-T (System of Observation and Experimentation, in the long term, for Environmental Research) dedicated to investigating the functioning of the forest ecosystem. The O₃HP site (680 m above mean sea level) is located 60 km north of Marseille and consists of a homogeneous 70-100 year-old coppice dominated by *Q. pubescens* (5 m in height; Leaf Area Index LAI = 2.2), which accounts for ≈ 90% of the biomass and ≈75% of the trees. A rainout-shelter above 300 m² of the canopy dynamically rainfall by deploying automated

shutters. This facility facilitated the study of *Q. pubescens* under natural and amplified drought, henceforth referred to as 'ND' and 'AD' plot respectively. In the present study, the device was deployed during rain events from the end of May until October 2012 in order to exclude 32% of the precipitation in the rain exclusion plot. In practice, almost all rainfall in late spring and summer was thus intercepted, intercepted increasing the number of dry days (<1 mm, Polade et al., 2014) by 22. This percentage corresponds with the highest IPCC projections made for the end of the century over the Mediterranean area, and accords with the precipitation reduction at $O_3HP$ during the driest years from 1967 to 2000 compared with the average precipitation over this period. Using an ombrothermic diagram ($P<2T$, with $P$=monthly precipitation in mm, and $T$=monthly air temperature in °C), we assessed that the summer 2012 drought period reaches 4.5 months in the AD plot, compared with 3 months in the ND plot. Ambient and soil environmental parameters were continuously monitored using a dense network of sensors (for details see Section 2.7). Access to the canopy was at two levels: $\approx 0.8$ and 3.5 m (top canopy branches) above ground level, with the highest level being the one at which we undertook this study. Further description can be found in Santonja et al. (2015).

## 2.2   Seasonal sampling strategy

Isoprene emission rate measurements were undertaken for at least one week per month from June 2012 to June 2013, except for the period from November 2012 until March 2013 when *Q. pubescent* is fully senescent with leaves remaining on the tree (marcescent species). This calendar enabled us to capture isoprene emissions during leaf maturity but also during bud break (April 2013) and just before leaf senescence (October 2012). Three trees were studied in each plot along the whole seasonal cycle, with a single branch at the top of the canopy predominantly sampled for each tree. More intensive measurements were carried out in June 2012 (3 weeks) and April 2013 when tree-to-tree and within-canopy variability was assessed. One ND branch was subsequently sampled throughout all intensive campaigns, and the five other ND and AD branches were alternately sampled during for one to two days (Genard-Zielinski et al., 2015). Isoprene samples were collected on cartridges packed with adsorbents, apart from in April 2013 when online isoprene measurements were conducted using a PTR-MS directly connected to the enclosure via a 50 m 1/4" PTFE line. When cartridges were used, samples (volume ranging between 0.45 L and 0.9 L, depending on the expected

emission intensity) were taken from sunrise to sunset, roughly every two hours. PTR-MS measurements allowed a higher sampling frequency (between 120-390 $s^{-1}$).

Branch enclosures were generally installed on the day before the first emission rate measurement was taken, and at least two hours beforehand in order for the plant to return to normal physiological functioning. Note that although senescence had just begun in October 2012, we did check that the enclosed branches were not senescent during these measurements.

## 2.3 Branch scale isoprene emissions and gas exchanges

Sampling was undertaken using two identical dynamic branch enclosures (detailed description in Genard-Zielinski et al., 2015). Briefly, the device consisted of a $\approx$ 60 L PTFE (PolyTetraFluoroEthylene) frame closed by a sealed, 50 µm-thick PTFE film, to which ambient air was introduced at $Q_0$ ranging between 11–14 L $min^{-1}$ using a PTFE pump (KNF N840.1.2FT.18®, Germany). Gas flow rates were controlled by mass flow controllers (Bronkhorst) and all tubing lines were PTFE-made. A PTFE propeller ensured the rapid mixing of air inside the chamber. The microclimate (PAR, $T$, relative humidity) inside the chamber was continuously monitored (relative humidity and temperature probe LI-COR 1400–04®, and quantum sensor LI-COR, PAR-SA 190®, Lincoln, NE, USA) and recorded (Licor 1400®; Lincoln, NE, USA). $CO_2/H_2O$ exchanges from the enclosed branches were also continuously measured using infrared gas analysers (IRGA 840A®, Licor) in order to assess the net assimilation $P_n$ (in $\mu mol_{CO2}$ $m^{-2}$ $s^{-1}$) and the stomatal conductance to water vapour $G_w$ ($mol_{H2O}$ $m^{-2}$ $s^{-1}$) using the equations from Von Caemmerer and Farquhar (1981) as detailed in Genard-Zielinski et al. (2015).

Total dry biomass matter (DM) was calculated by manually scanning every leaf of each sampled branch enclosed in the chamber and applying a dry leaf mass per area conversion factor (LMA) extrapolated from concomitant measurements made on the same site. The mean (range) DM was 0.16 (0.01 - 0.45) $g_{DM}$, and mean (range) LMA was 13.17 (0.82 - 36.67) $g_{DM}$ $cm^{-2}$.

Isoprene emission rates (ER) were calculated as:

$$ER = Q_0 \times (C_{out} - C_{in}) \times DM^{-1} , \tag{1}$$

where ER is expressed in $\mu gC$ $g_{DM}^{-1}$ $h^{-1}$, $Q_0$ is the flow rate of the air introduced into the chamber (L $h^{-1}$), $C_{in}$ and $C_{out}$ are the concentrations in the inflowing and outflowing air ($\mu gC$ $L^{-1}$), and DM is the sampled dry biomass matter ($g_{DM}$).

Throughout the seasonal cycle, except in April, isoprene was collected using packed cartridges (glass and stainless-steel) prefilled with Tenax TA and/or Carbotrap. Isoprene was then analysed in the laboratory according to a gas chromatography–mass spectrometry (GM-MS) procedure detailed in Genard-Zielinski et al. (2015), with a level of analytical precision greater than 7.5%.

In April 2013, two types of PTR-MS were used for online isoprene sampling and analysis. A quadrupole PTR-MS (HS-PTR-MS, Ionicon Analytik GmbH, Innsbruck Austria), connected to the ND branch enclosure, was operated at 2.2 mbar pressure, 60°C temperature and 500 V voltage in order to achieve an E/N ratio of $\approx$ 115 Td (E: electric field strength (V cm$^{-1}$); N: buffer gas number density (molecule cm$^{-3}$); 1Td=10$^{-17}$ V cm$^2$. The primary $H_3O^+$ ion count assessed at $m/z$ 21 was 3 10$^7$ cps, with a typically < 10% contribution monitored from the first water cluster ($m/z$ 37) and < 5% contribution from the $O_2^+$ ($m/z$ 32). Measurements were operated in scan mode ($m/z$ 21 to $m/z$ 210) every 380 s. After 15-20 min of sampling of incoming air, the outgoing air was sampled for 30 to 60 min. A high resolution (m/$\Delta$m $\approx$ 4000) time of flight PTR-MS (PTR-ToF-MS-8000, Ionicon Analytik GmbH, Innsbruck Austria) connected to the second enclosure used in our study enabled us to discriminate compounds when their masses differ at the tenth part. The main experimental characteristics were similar to the PTR-MS-Quad, but a voltage of 550 V was used in order to reach an E/N ratio of $\approx$ 125 Td. The $H_3O^+$ ion count assessed at $m/z$ 21 was 1.1 10$^6$ cps with a similar < 10% contribution monitored from the first water cluster (m/Z 37) and < 2.5% contribution from the $O_2^+$ ($m/z$ 32). The signal at m/z 69 corresponding to protonated isoprene was converted into mixing ratio by using a proton transfer rate constant k of 1.96.10$^{-9}$ cm$^3$.s$^{-1}$ (Cappellin et al., 2012), the reaction time in the drift tube, and the experimentally determined ion transmission efficiency. The relative ion transmission efficiencies of both instruments were assessed using a standard gas calibration mixture (TO-14A Aromatic Mix, Restek Corporation, Bellefonte, USA; 100 ± 10 ppb in nitrogen). Assuming an uncertainty of ±15% in the k-rate constants and in the mass transmission efficiency, the overall uncertainty of the concentration measurement is estimated to be of the order of ±20%. Background signal was obtained by passing air through a platinum catalytic converter heated at 300°C. Detection limits defined as three times the standard deviation on the background signal were 10 and 50 ppt with the PTR-ToF-MS and the HS-PTR-MS respectively. An intercomparison between both the cartridge+GC-MS and PTR-MS protocols was undertaken parallel with another

emitter present on the site (*Acer monspessulanum*); no significant difference was observed between the techniques (Genard-Zielinski et al., 2015).

The overall uncertainty (sampling + analysis) on ER assessment was between 20% and 25%.

## 2.4 Statistics

All statistics were performed on STATGRAPHICS® centurion XV by Statpoint, Inc. Differences in $P_n$, $G_w$, ER and *Q. pubescens* isoprene emission factors ($\varepsilon_{iso,Qp}$, see section 2.5 for details) between the ND and the AD plot were tested using Mann-Whitney U-tests. Seasonal changes in these ecophysiological parameters were tested using the Krusal-wallis test (K) and the analysis was performed separately on trees from the ND and AD plot.

Comparisons between COOPERATE environmental data (see Section 2.7) were made using a Wilcoxon test when data were not log-normal, and a *t*-test when log-normal.

## 2.5 Branch scale ER assessment using MEGAN2.1 emission model

Based on the latest version of the MEGAN model (MEGAN2.1, Guenther et al., 2012), *Q. pubescens* ER were assessed for the sampling conditions of our seasonal study using:

$$ER_{MEGAN} = \varepsilon_{iso,Qp} \, \chi_{Qp} \, \gamma_{iso} \qquad\qquad (2)$$

where :

- $\varepsilon_{iso,Qp}$ is the *Q. pubescens* isoprene emission factor calculated under each plot, every month of our study, as the slope of the linear regression between ER and $C_L \times C_L$ (see Section 3.2; in $\mu gC \ g_{DM}^{-1} \ h^{-1}$), where $C_L$ and $C_L$ are the instantaneous response of isoprene emissions to

photosynthetic photon flux density (PPFD) and T deviations to standard conditions (1000 $\mu mol \ m^{-2} \ s^{-1}$ and 30°C respectively) (Guenther et al., 1995); $C_L \times C_T$ were calculated using PAR and *T* recorded in the enclosure.

- $\chi_{Qp}$ is the fractional grid areal coverage taken equal to 1 since only *Q. pubescens* emissions (100%) were considered;

- and $\gamma_{iso}$ is the isoprene emission activity factor defined as:

$$\gamma_{iso} = \gamma_P \, \gamma_T \, \gamma_A \, \gamma_{SM} \, \gamma_C \qquad\qquad (3)$$

where:

- $\gamma_P$ and $\gamma_T$ are the isoprene empirical responses to light and temperature respectively, using instantaneous, daily and 10 days PPFD and T values (for details see Guenther et

al., 2012);

- $\gamma_A$ is the age emission activity based on empirical coefficients applied on new (0.05 applied for all April measurements), growing (0.6 for June) mature (1 for July and August) and old (0.6 for September and October) leaves;

- $\gamma_{SM}$, is the soil moisture dependence of isoprene emissions according to soil moisture value ($\theta$, m$^3$ m$^{-3}$) based on the Pegoraro et al. (2004) drought study on *Populus deltoides*:

$$\gamma_{SM} = 1 \qquad \text{for } \theta > \theta_1 \tag{4a}$$

$$\gamma_{SM} = (\theta - \theta_w)/\Delta\theta_1 \qquad \text{for } \theta_w < \theta < \theta_1 \tag{4b}$$

$$\gamma_{SM} = 0 \qquad \text{for } \theta_w < \theta_1 \tag{4c}$$

where $\theta_w$ is the wilting point (the soil moisture below which plants cannot extract water from soil, m$^3$ m$^{-3}$), $\Delta\theta_1 = 0.014$ is an empirical parameter, and $\theta_1 = \theta_w + \Delta\theta_1$. $\theta_w$ was assessed to be 0.15 m$^3$ m$^{-3}$ at the O$_3$HP, a value very close to the 0.138 m$^3$ m$^{-3}$ value given by Chen and Dudhia (2001) for clay and sand soil found at the O$_3$HP;

- and $\gamma_C$ is the CO$_2$ inhibition, set to 1 here as no CO$_2$ effect was tested in our study.

NB: in order to be comparable with our measurements carried out on top canopy leaves and expressed as net emission rates in the unit of µgC g$_{DM}^{-1}$ h$^{-1}$, no canopy environment coefficient C$_{CE}$ nor LAI was considered in the calculation of $\gamma_{iso}$, and thus in ER$_{MEGAN}$ (for further details see Guenther et al., 2012).

## 2.6    Branch scale ER assessment using an Artificial Neural Network trained on field data

The Artificial Neural Network (ANN) developed in this study to assess branch scale ER from *Q. pubescens* (henceforth referred to as *G*14) was based on a commercial version of the Netral NeuroOne software v.6.0 (inmodelia.com, France). The ANN was used as a Multi-Layer Perceptron (MLP) in order to calculate multiple non-linear regressions between a set of input regressors $x_i$ (the environmental variables measured at the O$_3$HP) and the output data (the measured isoprene ER). The assessed ER (ER$_{G14}$) was calculated as follows:

$$ER_{G14} = w_0 + \sum_{j=1}^{j=N}[w_{j,k} \times f(w_{0,j} + \sum_{i=1}^{i=n} w_{i,j} \times x_i)], \tag{5}$$

where $w_0$ is the connecting weight between the bias and the output, $N$ the number of neurons $N_j$, $f$ the transfer function, $w_{0,j}$ the connecting weight between the bias and the neuron $N_j$, $w_i$

the connecting weight between the input and the neuron $N_j$, and $x_i$ the $n$ input regressors. The MLP optimisation of the weights $w$ was achieved according to Boissard et al. (2008). Every input regressor $x_i$ was centrally-normalised. Two sub-datasets were considered, for the ND and AD plot respectively. For each sub-dataset, 80% of our data were used for training and optimising the MLP, and the remaining 20% were used for blind validation based on root mean square error (RMSE). Training/validation splitting was made using a Kullback-Liebler distance function available in NeuroOne v 6.0. Only the nonlinear hyperbolic tangent (*tanh*) function was tested as transfer function $f$. Up to $N=7$ neurons (distributed in only one layer) were tested for every ANN setting. Overtraining phenomenon (a too large number of neurons *vs* the number of input parameters) was checked against the $RMSE_{training}/RMSE_{validation}$ evolution *vs* the number $N$ of neurons tested: training was stopped for $RMSE_{training} > RMSE_{validation}$ when $N \geq 3$.

Among the other available statistical methods, ANNs present the advantage of being the most parsimonious, i.e., giving the smallest error for a same number of descriptors (see for instance Dreyfus et al., 2002). Moreover, the ANN approach, as is the case of other non-linear regression methods, is not particularly sensitive to regressors' co-linearity (Bishop, 1995; Dreyfus et al., 2002). On the other hand, one of the limitations of ANNs is that they can only be employed for interpolation within the range of values of the trained data, and not for extrapolation exercises beyond this range. Consequently, during the isoprene emission sensitivity to future climatic changes (see Section 2.8), only $x_i$ values fitting within the range of variation ($\pm 20\%$) tested during the training phase were considered; in total 21% of the data were thus rejected.

## 2.7 COOPERATE environmental database

Ambient and edaphic parameters used for the ANN optimization were obtained from the COOPERATE database (https://cooperate.obs-hp.fr/db) and daily averaged for each day of our study. Ambient PAR ($\mu$mol m$^{-2}$ s$^{-1}$) measured above the canopy at 6.5 m (Licor Li-190®; Lincoln, NE, USA) in the ND plot was used as the PAR reaching all of the top canopy branches studied. Ambient air temperature ($T$, °C) measured at 6.15 m (CS215, Campbell Scientific Ltd., UK) in the ND and AD plot was used for both sets of branches. Since some precipitation ($P$, mm) values were missing (<5%) from the COOPERATE database during our data processing, $P$ values from the nearby (< 10 km) Forcalquier meteorological station were used. The bias between cumulated $P$ ($P_{cum}$) curves at both sites was assessed and considered

in order to extrapolate the missing values at the O₃HP site. As $P$ was cumulated over 7, 14 and 21 days, the resulting bias was negligible ($\approx$ 1%) and no further adjustment was made. Soil water content (SW, L L⁻¹) and temperature (ST, °C) at -0.1 m (Hydra Probe II, Stevens, Water Monitoring Systems Inc., OR, USA) specific to each of the sampled trees were selected and extracted from the COOPERATE database; when soil data were missing, they were extrapolated from the nearest equivalent data point measurement. Daily mean PAR, $T$, $P$, SW and ST were cumulated over a time period ranging from 1 to 21 days before the measurement.

## 2.8 ORCHIDEE land surface model: providing future conditions to investigate ER sensitivity to climatic changes

Present-day $T$ and $P$ were assessed as the 2000-2010 daily averages derived from the ISI–MIP (Inter-Sectoral Impact Model Intercomparison Project) climate data set (Warszawski et al., 2014) over the Mediterranean area. This data set contains the bias-corrected daily simulation outputs of the earth system model HadGEM2-ES. Corresponding values for the 2090-2100 period were used to assess the expected range of future climatic changes. They were derived from two ISI–MIP future projections forced along two Representative Concentration Pathways (RCPs): the so-called 'peak-and-decline' greenhouse gas concentration scenario RCP2.6 (optimistic or moderate scenario), and the 'rising' greenhouse gas concentration scenario RCP8.5 (extreme or severe scenario). All $T$ and $P$ data were extracted for the entire Mediterranean region from the global ISI-MIP data set and subsequently averaged over the area.

Using these present and future $T$, $P$ and PAR values (ISI-MIP derived), the corresponding present and future SW and ST were assessed by running the global land-surface-model ORCHIDEE (ORganizing Carbon and Hydrology In Dynamic EcosystEms) over the European part of the Mediterranean region. The calculated SW and ST were averaged over this area. ORCHIDEE is a spatially explicit, process-based model that calculates the $CO_2$, $H_2O$ and heat fluxes between the land surface and the atmosphere. Vegetation species distributed at the Earth's surface are represented in ORCHIDEE through 13 Plant Functional Types (PFTs). Processes in the model are represented at the time step of ½ hour, but the variations of water and carbon pools are calculated on a daily basis. A detailed description of ORCHIDEE is provided by Krinner et al. (2005). Simulations over the European part of the Mediterranean region were performed with the ORCHIDEE model at 0.5 × 0.5° spatial resolution using the soil parameters (clay, silt and sand fractions) from Zobler (1986). Given

that this study focuses on isoprene emissions from *Q. pubescens*, we fixed the vegetation with the corresponding PFT 'temperate broad-leaf summer green tree'. The described ISI-MIP historical forcings and the ISI-MIP future projections were used as climate conditions for ORCHIDEE runs and ER assessment using $G14$. Equilibrium was reached by running ORCHIDEE on the first decade of the climate forcing (1961-1990) repeated in a loop, and the value of atmospheric $CO_2$ corresponding to the year 1961. Among the two different hydrology schemes available in ORCHIDEE, the physically based 11-layer scheme was used (Guimberteau et al., 2013).

ER sensitivity to moderate and severe temperature and/or precipitation changes was evaluated using $G14$ under 6 cases: (i) the 'T' (respectively, 'P') test was conducted considering only $T$ and ST (respectively, only $P$ and SW) changes according to RCP2.6 scenario; (ii) the 'TT' and 'PP' tests were similar to the 'T' and 'P' tests but considered changes according to RCP8.5 scenario; (iii) the 'T+P' (respectively, 'TT+PP') test combined the effect of $T$, ST, $P$ and SW changes according to RCP2.6 (respectively, RCP8.5).

## 3    Results

### 3.1    Environmental conditions observed at the O₃HP

Mean daily ambient air temperature $T$ varied between -3 and 26°C (January 2013 and August 2012 respectively, Fig. 1a). Seasonal PAR variations were in line with $T$ variations, with the daily mean peaking at 900 µmol m$^{-2}$ s$^{-1}$ in July (Fig. 1b). In 2012, the amplification of the ND was adjusted from May to reach its maximum (32%) in July and maintained until November when rain exclusion was stopped (Fig. 1c). The annual $P_{cum}$ in the AD plot was lower by 273 mm than in the ND plot at the end of 2012 (782 compared to 509 mm). In 2013 the AD started only at the end of June, simulating a later amplification. From August until October 2012, SW was 50-90% lower in the AD plot than in the ND plot ($\approx$ 0.02 and to 0.05 L$_{H2O}$ L$_{soil}$$^{-1}$ respectively in August, Fig. 1d). The AD plot soil water deficit remained significant until the end of the experiment (Mann-Whitney, $P<0.05$ in June 2012, $P<0.001$ from July 2012 to June 2013), although the rain exclusion system was not activated between December 2012 and June 2013.

No significant difference was noticed for monthly PAR and $T$ means between the ND and the AD plot, except in September 2012 when branches sampled on the ND plot received significantly more PAR than branches on the AD plot (Mann-Whitney, $P<0.001$). This

difference could be due to an orientation of the branches sampled in the ND plot in September that enabled greater receipt of PAR during our measurements than the AN sampled branches.

### 3.2 Gas exchange and isoprene seasonal variations

$G_w$ and $P_n$ showed similar seasonal patterns in both plots (Figs. 2a, 2b), with the lowest values in July-September (10-20 $mol_{H2O}$ m$^{-2}$ s$^{-1}$ and $\approx$1 $\mu mol_{CO2}$ m$^{-2}$ s$^{-1}$ respectively), and the highest in June (80-170 $mol_{H2O}$ m$^{2}$ s$^{-1}$ and $\approx$9 $\mu mol_{CO2}$ m$^{-2}$ s$^{-1}$ respectively). Respiration dominated over gross $CO_2$ assimilation in April, resulting in negative net assimilation ($P_n \approx$ -1 $\mu mol_{CO2}$ m$^{-2}$ s$^{-1}$) in both plots. In contrast, $G_w$ and $P_n$ were not influenced by water stress in the same way. Whereas $G_w$ was significantly reduced under AD from July 2012, $P_n$ remained stable, except in June 2013 when $P_n$ values that were twice as high under AD than ND were observed. It is important to note that the tomography measurements made at this site showed that oak roots were predominantly distributed in the outermost humiferous horizon located above a calcareous slab at a 10-20 cm depth, and that only very few roots crossed this slab.

Water stress only affected the ER seasonal pattern during summer (Fig. 2c). Maximum ER was delayed by a month in the AD plot (104.8 $\mu gC$ $g_{DM}^{-1}$ h$^{-1}$ in August) in comparison to the ND plot (78.5 $\mu gC$ $g_{DM}^{-1}$ h$^{-1}$ in July). ER was lowest in October ($\approx$ 6 $\mu gC$ $g_{DM}^{-1}$ h$^{-1}$ in both plots). During April bud-break and isoprene emission onset, ER was as low as 0.5 and 1 $\mu gC$ $g_{DM}^{-1}$ h$^{-1}$ in the ND and AD plot respectively.

Although $\varepsilon_{iso,Qp}$ was calculated every month as the slope of ER $vs$ $C_L \times C_T$ (as in Guenther et al., 1995), this correlation was not significant in July, especially in the case of AD branches (P>0.05, R$^2$ = 0.06 and 0.01 for ND and AD respectively). As a result, $\varepsilon_{iso,Qp}$ in July was calculated by averaging ER measured under environmental conditions close to 1000$\pm$100 $\mu mol$ m$^{-2}$ s$^{-1}$ and 30$\pm$1°C. In general, AD branches showed poorer ER $vs$ $C_L \times C_T$ correlations than branches growing in the ND plot (data not shown). $\varepsilon_{iso,Qp}$ was significantly higher by a factor of 2 in August and September for the AD branches compared to the ND (Fig. 2d). As for ER, $\varepsilon_{iso,Qp}$ maximum was reached in August (137.8 $\mu gC$ $g_{DM}^{-1}$ h$^{-1}$) in the AD plot, while the maximum in the ND plot occurred in July (74.3 $\mu gC$ $g_{DM}^{-1}$ h$^{-1}$). The general high variability observed in April during the isoprene emission onset (some branches were already emitting, while some were not yet emitting isoprene, regardless of their locations in the AD/ND plots) was as large as the AD-ND variability, and thus could not solely be attributed

to the water stress treatment. The relative annual $\varepsilon_{iso,Qp}$ difference between ND and AD was +45%.

### 3.3 Modeling the isoprene seasonal variations of *Q. pubescens* at the O₃HP

Given that we were aiming to test the capacity of an empirically based isoprene emission model to describe seasonal ER variability and sensitivity to drought observed during this study, we tested the latest version of the MEGAN model, which is widely used for air quality and climate change applications (MEGAN2.1, Guenther et al., 2012). In particular, the ability of its soil moisture coefficient activity $\gamma_{SM}$ (Eq. 4a-4c) to assess the observed effect of ND and AD treatments was examined over wilting point $\theta_w$ values ranging from 0.01 to 0.15 $m^3$ $m^{-3}$, representative of a large brand of soils (Ghanbarian-Alavijeh and Millàn, 2009). Indeed, Müller et al., (2008) showed that isoprene assessments were very sensitive to $\theta_w$. For the record, $\theta_w$ was 0.15 $m^3$ $m^{-3}$ at the O₃HP.

Assessed (ER$_{MEGAN}$) and observed (ER) isoprene emission rates were compared separately for ND and AD. However, given that the rainout shelter was implemented close to the commencement of our study in June 2012, measurements carried out in the AD plot were not distinguished, only in the case of this month, from the ones taken in the ND plot (AD and ND data were thus mixed for June 2012).

For $\theta_w$<0.05 $m^3$ $m^{-3}$, and regardless of the $\theta_w$ value, MEGAN2.1 captured more than 80% of the ER variability in the ND plot (y=0.15x$^{1,5}$, R² =0.81, Fig. 3a), but less ($\approx$ 50%) in the AD plot (R²=0.53 and 0.54 for $\theta_w$=0.005 and 0.01 $m^3$ $m^{-3}$ respectively, Fig 3b). An overall over-estimation of 25% was associated with the MEGAN2.1 assessment for both treatments. On the contrary, for $\theta_w \geq 0.05$ $m^3$ $m^{-3}$, most of the isoprene emissions were set to zero by MEGAN2.1 in the AD plot, while in the ND only June observations were correctly assessed with an overall over-estimation (whatever the $\theta_w$ values) of $\approx$ 10% (R² ranging from 0.76 to 0.80 for $\theta_w$=0.15 and 0.1 $m^3$ $m^{-3}$ respectively). If some of the July ER$_{MEGAN}$ were fairly close to the observations for $\theta_w$=0.1 $m^3$ $m^{-3}$, the overall correlation was poor (y=0.2x+49.5, R²=0,02).

Assuming that the discrepancies between ER$_{MEGAN}$ and ER only resulted from the $\gamma_{SM}$ formulation in MEGAN2.1 (and not from the other activity coefficients $\gamma_P$, $\gamma_T$ or $\gamma_A$ used, Eq. (3)), ER/ER$_{MEGAN}$ was calculated for both ND and AD treatments and was considered against the measured SW. In the ND treatment, ER/ER$_{MEGAN}$ was not found to be significantly

dependent on SW ($y=0.653e^{10.52x}$, $R^2=0.13$, Fig. 4a). However, in the AD plot, $ER/ER_{MEGAN}$ increased exponentially with SW ($y = 0.192e^{51.93x}$, $R^2=0.66$, Fig. 4b), and in particular when SW became higher than the wilting point $\theta_w$ measured at the O₃HP site (0.15 m³ m⁻³). Similar findings were obtained for SW-7, SW-14 and SW-21, for both the ND and AD treatments (Table 1).

In order to provide a better description of the impacts of ND and AD on ER as observed at the O₃HP, an empirical type model, based on ANN optimisation of our observations at the O₃HP, was developed specifically for *Q. pubescens* isoprene emissions. Training and validation of the different ANNs tested were made using values of ER, *T*, *P*, PAR, ST and SW measured at the O₃HP (COOPERATE database). Environmental regressors $x_i$ were integrated, using daily means, over a period ranging from 0 to 21 days prior to the measurements.

Among the different ANN settings tested, an optimised architecture, *G*14 (lowest RMSE between calculated and measured values, no overtraining, best correlation between measured and calculated ER over the whole range of value, see Boissard et al., 2008) was found for *N*=3 and a set of 16 $x_i$ with their corresponding connecting weights $w_i$ (Appendix 1). The final optimised RMSE (validation data) was 8.5 µgC $g_{DM}^{-1}$ h⁻¹, for ER values ranging from 0.06 to 113 µgC $g_{DM}^{-1}$ h⁻¹, and represents 35% of the mean (22.7 µgC $g_{DM}^{-1}$ h⁻¹). More than 80% of the ER seasonal variations were assessed by *G*14, whatever the water treatment (ND or AD) and the month, except in July (Fig. 5a) when ER variability was always poorly represented whatever the different ANN settings considered; these July data correspond to the period where trees were started to adapt to ND and AD, and were possibility not enough represented in our dataset to be well taken into account by our statistical approach. An overall underestimation of 6% and 12% was observed in the ND and AD respectively. For comparison, $ER_{MEGAN}$ calculated with a value $\theta_w$ of 0.15 m³ m⁻³ are presented again in Fig. 5b for both the ND and AD treatment.

Under ND, the global contribution of the two lowest frequencies (-14 and -21 days) considered in *G*14, was, relative to the contribution of the two highest frequencies (instantaneous and -7 days), higher than under AD (Fig. 6). In particular, in October 2012 and April and June 2013, the two lowest frequencies respectively represented 20, 97 and 50% of the total in the ND compared to 3%, 55% and 26% in the AD.

**3.4 ER sensitivity to expected climatic changes over the European Mediterranean area**

Present and future $T$, $P$ and PAR (ISI-MIP-derived), and SW and ST (ORCHIDEE-derived) were integrated over periods ranging from 0 to 21 days in order to be used in $G14$ and to assess $ER_{G14}$ for present and future cases. Moderate (respectively, severe) changes with regard to the present of SW, $P$, ST, $T$ and PAR were additionally calculated according to RCP2.6 (respectively, RCP8.5) scenario; however, PAR relative changes were not considered as they were negligible for both moderate and severe scenarios.

Moderate changes of the environmental conditions (RCP2.6 scenario) implied a systematic positive monthly $\Delta T$ throughout the year, whereas $\Delta P$ was found to be positive only during the winter and negative during the summer (Fig. 7a). ST and SW changes were found to be in line with $T$ and $P$ respectively. The highest monthly relative changes were for $P$ (+75% in February and -30% in July), whereas the smallest were for SW. Monthly ST and $T$ relative changes remained more or less constant (between +7% and +10%) between February and November. Overall, $T$ and $P_{cum}$ absolute (relative) annual changes were +1.4°C and +34 mm respectively (+9.1% and +4.8% respectively, Table 2).

Under more severe environmental changes (RCP8.5 scenario), monthly $T$ and ST increased all year round, whereas $P$ and SW generally decreased, except in January, February and November, when relative $P$ changes were negligible (Fig. 7b). The annual absolute (relative) changes for $T$ and $P_{cum}$ were +5.3°C and -124 mm respectively (+34% and -24%, respectively, Table 2). In these conditions, the annual $\Delta P_{cum}/P_{cum}$ was similar to the reduction experienced at the O₃HP during our study (-30%). The highest monthly relative changes were found for ST: +96% and +86% in January and December respectively. During summertime the highest relative changes were found for $P$ (-55% and -62% in July and August respectively).

$ER_{G14}$ was found to systematically increase compared to the present under 'T' and 'TT' changes, with an annual relative change of +80% and +240% respectively (Fig. 8a). The highest relative changes were noted in June and July. In contrast, $ER_{G14}$ was almost not sensitive to 'P' or 'PP' changes, whatever the month (annual relative change of +10% and +8% respectively, Fig. 8b). When the combined impacts of changes in temperature and precipitation were considered, $ER_{G14}$ was found to systemically increase all year round, following a seasonal trend that was extremely close to that found for the 'T' and 'TT' tests (Fig. 8c). However, the additional effect of the precipitation changes enhanced the increase noticed for temperature changes only: the annual increase was +100% ('T+P') and +280%

('TT+PP') compared to +80% ('T') and +240% ('TT'). Note that the $ER_{G14}$ seasonal trend calculated for the present did not match our observed ER variations. Indeed $ER_{G14}$ was tuned using environmental parameters averaged over 24 hours (and therefore integrated over the daytime and night-time period), and were thus much lower than the environmental parameters measured during our daytime-only samplings (especially for PAR and $T$).

## 4    Discussion

### 4.1    Impact of water stress on seasonal gas exchanges and isoprene emission of *Q. pubescens*

In spite of a significant $G_w$ reduction in summer 2012 owing to the AD, *Q. pubescens* maintained a positive $P_n$ during the summer, regardless of water stress (ND or AD). Electric resistivity tomography measurements carried out on the $O_3HP$ site revealed the heterogeneity of the karstic substrate, organised as soil pockets developed between limestone rocks. Water and nutrient pools and dynamics probably differed greatly between the shallow upper soil layers and the soil pockets developed between limestone rocks. However, the soil trenches in the site revealed that a calcareous slab often developed at a depth of 10-20 cm and that the roots of the oaks were often distributed in this humiferous horizon close to the surface, with very few roots crossing this slab. Water supply from layers deeper than 10-20 cm was thus not considered. Such behaviour enables trees to limit evapotranspiration under water stress, and as a drought-acclimated species permits them to ensure sufficient accumulation of carbohydrates for the winter (Chaves et al., 2002). Such strategy was also observed in a study conducted on the same species but under greenhouse conditions (Genard-Zielinski et al., 2015). The seasonal regulation/conservation of $P_n$ and $G_w$ enabled isoprene emissions to be maintained even during the summer water stress (ND and AD).

The maximum $\varepsilon_{iso,Qp}$ in both plots was close to previously measured values obtained for the same species under Mediterranean conditions during greenhouse and *in-situ* experiments (114.3 and 134.7 $\mu gC.g_{DM}^{-1}.h^{-1}$) by Genard-Zielinski et al. (2015) and Simon et al. (2005) respectively. The difference observed in April 2013 between $\varepsilon_{iso,Qp}$ in the ND and AD could not be attributed solely to the AD effect. Indeed, apart from a possible 'memory effect' of the AD applied during 2012, the observed difference was probably due to the high natural variability in bud-breaking and isoprene emission onset at this point of the year. The observed significant increase (a factor of 2) of $\varepsilon_{iso,Qp}$ under AD (August and September) illustrates how

isoprene is likely to be important for short-term *Q. pubescens* drought-resistance, in particular through the ability of isoprene to stabilise the thylakoids membrane, under (for example) thermal or oxidative stress (Peñuelas et al. 2005; Velikova et al. 2012). Moreover, previous studies have highlighted the possibility for a plant growing under water stress to synthesise isoprene using an alternative carbon source (extra-chloroplastic carbohydrates) (Lichtenthaler et al., 1997; Funk et al., 2004; Brili et al., 2007). For species emitting other BVOC than isoprene, but studied in the Mediterranean area under water stress, Lavoir et al. (2009) reported lower (a factor of ≈2) monoterpene emission rates from *Quercus ilex* under AD from June to August, during the 2nd and 3rd year of rain exclusion. Since *Q. ilex* does not possess specific leaf reservoirs for monoterpene storage, *Q. ilex* monoterpene emissions are hence *de novo* and their emissions are tightly related to their synthesis according to light and temperature as isoprene.

The significant uncoupling between ER and $C_L \times C_T$ reported for the July measurements occurred when SW significantly decreased to their seasonal minimum values (0.05 and 0.03 $m^3$ $m^{-3}$) at the $O_3HP$ in both plots. A similar uncoupling has also been observed for some other strong isoprene emitters under water stress (*Quercus serrata, Quercus crispula*, Tani et al., 2011). These findings may confirm these authors' assumptions that extra-chloroplastic isoprene precursors supply the carbon basis for isoprene biosynthesis (and not only from $CO_2$ fixed instantaneous in the chloroplast) when water stress occurs, which explains why isoprene emissions become less dependent on the classical abiotic factors PAR and *T* as considered by Guenther et al. (1995).

## 4.2 Improving consideration of the drought effect in isoprene emission models

Since ND and AD conditions tested by *Q. pubescens* in our study stood aside from optimal growth conditions under which empirical emission models perform fairly well, it was interesting to test the ability of MEGAN2.1 to reproduce the observed impacts of a water deficit, as in $O_3HP$, on isoprene emissions. The formulation of the MEGAN2.1 soil moisture factor $\gamma_{SM}$, wilting point-centred, was deemed inadequate to reproducing the observed isoprene variability of a drought-adapted emitter such as *Q. pubescens*. Thus, MEGAN2.1 very successfully reproduced observed ER variability under the ND (more than 80%) only when $\gamma_{SM}$ was not operating; in fact, only when very low values of the wilting point were selected ($\theta_w \leq 0.01$ $m^3$ $m^{-3}$), $\gamma_{SM}$ was set to 1. In practice, wilting point values lower than 0.01 $m^3$ $m^{-3}$ are encountered very rarely, and only for loamy sand soils (Ghanbarian-Alavijeh and

Millàn, 2009), and so did not apply in the case of *Q. pubescens* in the present study. Once higher $\theta_w$ ($\geq 0.05$ $m^3$ $m^{-3}$) were tested, $\gamma_{SM}$, and with it almost all the isoprene emissions, rapidly decreased to zero once the drought was underway (i.e., after the June measurements). On a larger scale (over the subtropical Africa), Müller et al. (2008) found that MEGAN underestimation of isoprene emissions were also the largest after the drought was reached. Consequently, for a drought-adapted isoprene emitter, not only was the wilting point not found to be a relevant parameter to be considered in the expression of $\gamma_{SM}$, but also a formulation that could stop isoprene emissions, whatever the drought intensity.

The fact that under ND the discrepancies between $ER_{MEGAN}$ and ER were not found to be contingent on the soil water content SW (Fig. 4a) illustrates that under a natural drought intensity, the capacity of a drought-resistant species to emit isoprene, that is to trigger physiological regulations to protect its cellular structures, is primarily due to its natural adaptation, and not to the water available in the soil. Isoprene emissions became SW dependent only when the adaptation of *Q. pubescens* to its 'natural' environment was threatened (i.e., the AD treatment, Fig 4b). Thus, for a species that is not adapted to drought, such as *Populus deltoides*, the appearance of unusual water stress conditions would strongly affect and limit its isoprene emissions, as previously reported by Pegoraro et al. (2004). Indeed, this reference is the only one used by Guenther et al. (2006) to account for the impact of the soil water content in MEGAN2.1; the $\gamma_{SM}$ factor cannot effectively account for isoprene emission variability for drought-adapted emitters such as *Q. pubescens*. Such a discrepancy under conditions other than Mediterranean was also noticed by Potosnak et al. (2014) during a seasonal study over a mixed broad-leaf forest primarily composed of *Q. alba* L. and *Q. velutina* Lam. (Missouri, USA). Guenther et al. (2013) have suggested that including the soil moisture, averaged over longer periods of time (such as the previous month and not only the mean over the previous 240 hours) may help to improve predictions during drought periods. In this study we found that the discrepancies between $ER_{MEGAN}$ and ER were not related to the frequency over which SW was considered (Table 1): under ND they remained SW independent, whereas under AD the correlation between $ER/ER_{MEGAN}$ and SW remained of the same order ($0.66 \leq R^2 \leq 0.38$), but with a best fit found for the soil water content of the current day. These findings suggest that the formulation of the soil moisture activity factor could be improved in MEGAN2.1 if at least two distinct types of isoprene emitters were considered: (i) non-drought-adapted species (such as *Populus deltoides*) from which isoprene emissions would be modulated using the actual $\gamma_{SM}$ formulation; (ii) drought-adapted emitters

(such as *Q. pubescens*), for which $\gamma_{SM}$ would modulate isoprene emissions relative to SW, without diminishing them to zero, in an exponential way similar to the expression found in this study: $\gamma_{SM} = 0.192e^{51.93\,SW}$ (see Section 3.3). However, validation of such an expression to other drought-adapted isoprene emitters, as well as to other drought-adapted BVOC emitters, is required and will necessitate further field/controlled *ad hoc* experiments.

Moreover, the largest discrepancies between $ER_{MEGAN}$ and ER were noticed for the measurements in April and for some of those in June (Figs. 3 and 4), i.e., in periods when the drought (whether natural or amplified) was yet to be completely underway during our study. This highlights that ER variability during the onset and seasonal increase of isoprene emissions was not solely drought- or SW-dependent, even in a water-limited environment such as the O₃HP. Indeed, as observed for *Q. alba* and *Q. macrocarpa* Michx, the isoprene onset was found to be strongly correlated with ambient temperature cumulated over ≈2 weeks (200 to 300 degree day, Dd, °C), while the maximum ER was observed at 600-700 Dd °C (respectively Geron et al., 2000 and Petron et al., 2001). However, if part of this dynamical regulation is already included in MEGAN2.1 through its emission activity factors $\gamma_T$ and $\gamma_A$ (see Eq. 3), the combined effect of temperature regulation and drought is not fully accounted. For instance, Wiberley et al. (2005) observed that the onset of kudzu isoprene emissions was shortened by one week under elevated temperature compared to cold growth. $ER_{G14}$ consequently became more sensitive to rapid environmental changes as drought intensity increased: the overall averaged relative contributions of the regressors $x_i$ cumulated over 14 and 21 days decreased by 45% and 29% in the ND and AD respectively. Interestingly, these changes were found to be highest during the months of October 2012 (35% and 8% in the ND and AD respectively), April 2013 (from 96% to 55% in the ND and AD respectively) and June 2013 (49% and 26% in the ND and AD respectively, Fig. 6). Therefore, during the senescence and onset periods, the drought affected the dynamical regulation of isoprene emission more than the emissions themselves. Thus, an ANN approach as used in this study to develop $G14$ highlights the importance of including a modulation along the season of the range of frequencies over which the relevant environment regressors should be considered.

### 4.3    How will climatic changes affect the seasonal variations of *Q. pubescens* isoprene emissions in the Mediterranean area?

In the future, the Mediterranean area investigated in this study will face changes in terms of precipitation regime (thus of soil water content), and/or changes in ambient temperature (thus

of soil temperature). Depending on the $CO_2$ trajectory scenario considered, the annual $P_{cum}$ would remain more or less stable (RCP2.6), or decrease by 24% (RCP8.5); however, the seasonal regime would change, with a summer reduction of $P$ in both cases. The $O_3HP$ experimental strategy used in this work illustrates the upper limit of the drought intensity that *Q. pubescens* could undergo by 2100 in the Mediterranean area. On the other hand, temperature would increase regardless of the scenario and month, from 1.4 (+10%) to 5.3°C (+34%) annually.

As expected, $ER_{G14}$ was found to increase appreciably with temperature increase, from 80% annually in the RCP2.6 scenario to 240% in RCP8.5 (Fig. 8a). If such an increase is generally estimated and observed when considering a range of temperature enhancements that accord with future projected changes (Peñuelas and Staudt, 2010), such a response seems fairly unclear under Mediterranean water deficit conditions (Llusià et al., 2008, 2009). On a global scale, Müller et al. (2008) estimated a 20% decrease of isoprene due to soil water stress. In our case, isoprene emissions were found to be scarcely sensitive to $P$, whatever the intensity of changes: at most, annual $P$ would increase isoprene emissions by 10%, regardless of the intensity of $P$ changes investigated over the scenario considered (Fig. 8b). This finding is in line with our observations: except in October 2012, monthly averaged ER were not significantly different in the ND and the AD (Fig.2c). However, if the observed SW did differ between the ND and the AD plots ($\approx$ a factor of 2, Fig. 1), SW calculated by the ORCHIDEE model was almost entirely unaffected by the $P$ changes, even in the severe scenario RCP8.5. Such an uncoupling between $P$ and SW could be explained by modifications in the ORCHIDEE model of the overall soil water evapotranspiration, runoff and drainage which in short lead to near-constant SW values. In order to test the impact of the sole SW changes within a similar range to that observed at the $O_3HP$ between ND and AD, $ER_{G14}$ seasonal variation was calculated using present SW multiplied every day by 0.5, 0.75, 1.5 and 2 (Fig. 9). Surprisingly, $ER_{G14}$ was almost unchanged when SW was reduced (-2% and -13% annually for 0.5×SW and 0.75×SW respectively). $ER_{G14}$ increased only when SW increased: +51% and +93% annually for 1.5×SW and 2×SW respectively. These results are in line with our findings that, under a certain level of SW, isoprene emissions from a drought-adapted emitter such as *Q. pubescens* are no more affected by soil water content. Indeed, under ND, $ER/ER_{MEGAN}$ was not correlated with SW, but under AD, $ER/ER_{MEGAN}$ remained more or less stable when SW was lower than the wilting point (Fig. 4 and Section 3.3). Isoprene emission variations would be highly SW-dependent only for the highest SW values: (i) in the spring and in the beginning of the summer when the drought is not completely underway; (ii) in the

fall when the drought stress is fading away and when the highest differences are assessed between $ER_{G14}$ calculated for SW-present and for 2×SW (Fig. 9). When the $T$ and $P$ effects were combined, the seasonal variation of $ER_{G14}$ was affected in a similar way to when the sole $T$ effect was considered, but with an enhanced increase: +20% and +40% between 'T'

and 'T+P' tests, and between the 'T+P' and 'TT+PP' tests respectively (Figs. 8a and 8c). Such higher sensitivity of *Q. pubescens* isoprene emissions to temperature stress under drought was also observed by Genard-Zielinski (2014). Understandably, the $G14$ algorithm developed in this study to assess isoprene emissions in future climates should be validated through a longer period of measurement, in order to assess how *Q. pubescens* acclimates over

a more extensive period of drought, and to confirm or deny these findings. In this context, measurements have been carried on at the $O_3HP$ on the same branches as the ones studied in this work since June 2013 (Saunier et al., 2017).

These findings were attained considering an unchanged *Q. pubescens* biomass, i.e., unaffected by long-term acclimation to $T$ and drought increase. However, one can question whether *Q.*

*pubescens* could maintain such a high allocation of its primary assimilated carbon (primary plant metabolites, PPMs) to isoprene emissions (secondary plant metabolites, PSMs). Indeed, Genard-Zielinski et al. (2015) have shown that under moderate and severe drought, *Q. pubescens'* aerial and foliar growth is negatively affected. Thus, in the long term, such a cost of drought could affect the overall energy budget and expedite plant senescence (Loreto and

Schnitzler, 2010). The assessed $ER_{G14}$ increase could then be offset or even reversed.

On the other hand, one should also consider the additional co-effects of the $CO_2$ increase expected in the future. Bytnerowicz et al. (2007) have reported that if temperature increase proves to have little effect, elevated $CO_2$ would favour both the growth and water use efficiency of plants, and account for a 15-20% increase in forest NPP. When $CO_2$

enhancement was considered, the leaf mass per square metre of the PFT tested in ORCHIDEE in this study (broad-leaf temperate) was predicted to undergo a relative increase by 35% and 100 % under RCP2.6 and RCP8.5 respectively. Tognetti et al. (1998) observed a similar positive effect on the assimilation rate of both *Q. pubescens* and *Q. ilex* during a long-term $CO_2$ enhancement study, and measured a net increase in the diurnal course of isoprene

emissions. Thus, the major impact of future climate change on isoprene emissions could eventually be related to a general change in land cover, with Mediterranean species shifting to more favourable conditions.

# 5    Conclusion

The study carried out in 2012-2013 at the O₃HP on *Q. pubescens* was the first to test *in natura* and on a seasonal scale the effects of drought (ND and AD) on gas exchange, and in particular isoprene emissions of a mature coppice. This unique set of experimental data has confirmed how a drought-adapted species was able (i) to limit its evapotranspiration under water stress, even in summer, in order to maintain a similar level of net assimilation whatever the drought intensity and (ii) to emit similar or even higher amounts of isoprene in order to protect cellular structures under drought (ND or AD) episodes. In an environment such as the O₃HP (elevated ambient temperature and scarcity of the water available), and for a drought-adapted emitter such as *Q. pubescens*, isoprene emissions were thus maintained, and in the ND, their variability was not dependent on the soil water content. However, under the AD treatment, isoprene emissions were found to exponentially decrease with SW, in particular when SW was lower than the wilting point measured at the site ($\theta$w = 0.15 m³ m⁻³).

Since the intensity of isoprene emissions in the Mediterranean area is large, and can occur together and close to large urban emissions of other reactive compounds (in particular NOx emissions), the impacts of future environmental changes on isoprene emissions in this area need to be assessed as precisely as possible. The latest version of the empirical isoprene model, MEGAN2.1 was found to be unable to reproduce the effect of drought on isoprene emissions from *Q. pubescens*, regardless of the drought intensity (ND or AD). However, for such a drought-adapted emitter, MEGAN2.1 performed very well to capture the seasonal ER variability (more than 80%) under ND when its soil moisture activity factor γ$_{SM}$ was not operating (γ$_{SM}$=1); this performance decreased to ≈50% in the AD treatment. We suggest that, in addition to the actual γ$_{SM}$ expression, which is only valid for non-drought-adapted emitters, a specific formulation should be considered for drought-adapted emitters involving an exponential decrease of isoprene emission with SW decreasing to above-zero values, as proposed in this study for *Q. pubescens*. An ANN approach similar to that undertaken to develop *G*14 highlighted its ability to extract from appropriate field data measurements the relevant environmental regressors to be considered and the relevant frequency over which they should be employed. *G*14 was able to reproduce more than 80% of the ER seasonal variability observed for *Q. pubescens*, whatever the drought intensity. Moreover the application of *G*14 to future climate environmental data derived from IPCC RCP2.6 and RCP8.5 scenarios suggests that isoprene emissions in the future will be mainly affected by warmer conditions (up to an annual 240% increase for the most severe warming scenario), not

by drier conditions (at most, a 10% increase annually). The major impact of amplified drought will actually consist of enhancing (by up to 40%) the sensitivity of isoprene emissions to thermal stress.

*Acknowledgements.* We are particularly grateful to P. E. Blanc, J. C. Brunel G. Castagnoli, A. Rotereau and other OHP staff for support before and during the different campaigns. We thank members of the DFME team from IMBE: S. Greff, C. Lecareux, S. Dupouyet and A. Bousquet-Melou for their help during measurements and analysis. This work was supported by the French National Agency for Research (ANR) through the projects CANOPÉE (ANR-2010 JCJC 603 01) and SecPriMe$^2$ (ANR-12-BSV7-0016-01), INSU (ChARMEx), CNRS National program EC2CO-BIOEFECT (ICRAM project), CEA. We are grateful to ADEME/PACA for PhD funding. For O$_3$HP facilities, the authors thank the research federation ECCOREV FR3098 and the LABEX OT-Med (no. ANR-11-LABEX-0061), funded by the French Government through the A\*MIDEX project (no. ANR-11-IDEX-0001-02). The authors thank the MASSALYA instrumental platform (Aix Marseille Université, lce.univ-amu.fr) for the analysis and measurements used in this publication

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

**Figure caption**

Figure 1: Seasonal variations of daily environmental parameters measured at the $O_3HP$ from

March 2012 to June 2013. (**a**) Ambient air temperature $T$ was obtained at 6.5 m above ground level (a.g.l.), approximatively 1.5 m above the canopy. (**b**) Photosynthetic active radiations PAR received at 6.5 m a.g.l. in the ND plot. (**c**) Cumulated precipitation $P_{cum}$ measured over the ND (blue) and AD (red) plot. (**d**) Mean soil water content SW ± SD measured at -0.1 m depth from various soil probes in the ND (blue, $n=3$) and AD (red, $n=5$) plot.

Figure 2: Seasonal variations of monthly *Q. pubescens* gas exchanges observed at $O_3HP$ (June 2012 to June 2013) under ND (blue) and AD (red) (mean ± SD). (**a**) Stomatal conductance to water vapour $G_w$. (**b**) Net photosynthetic assimilation $P_n$. (**c**) Measured branch isoprene emission rate ER. (d) Isoprene emission factor ($I_s$) calculated according to Guenther et al. (1995) using *in situ* ER *vs* $C_L \times C_T$ correlations, except in July where mean ER measured under enclosure conditions close to 1000 µmol m$^{-2}$ s$^{-1}$ and 30 °C was used. Differences between ND and AD using Mann-Whitney U-tests are denoted using lower case letters (a>b>c>d). Differences among water treatment stress using Kruskal-wallis tests are denoted by asterisks (*: P<0.05; **: P<0.01; ***: P<0.001).

Figure 3: Comparison between isoprene emission rates (in µgC $g_{DM}^{-1}$ h$^{-1}$) calculated using MEGAN2.1 ($ER_{MEGAN}$, Guenther et al., 2012) and measured isoprene emission rates (ER) versus the wilting point value $\theta w$ (0.005 to 0.15 m$^3$ m$^{-3}$), from June 2012 to June 2013, under (**a**) ND (*n*=267) and (**b**) AD (*n*=138). Since the rain exclusion device was only implemented soon prior to our study's commencement in June 2012, the ND and AD measurements were considered together for June 2012. Linear regressions for ND June 2012 were: y=1.13x-12.05, R²=0.80 ($\theta w$=0.05 m$^3$ m$^{-3}$); y=1.13x-7.13, R²=0.80 ($\theta w$=0.1 m$^3$ m$^{-3}$); y=1.12x-16.94, R²=0.76 ($\theta w$=0.15 m$^3$ m$^{-3}$). The dotted line is the 1:1 line.

Figure 4: Ratio between observed (ER) and calculated ($ER_{MEGAN}$) isoprene emission rates versus the soil water content SW measured at the $O_3HP$, under (**a**) ND (*n*=267) and, (**b**) AD (*n*=138). Given that the rain exclusion device was only implemented just before our study began in June 2012, the ND and AD measurements were considered together for June 2012. The dotted line is for SW=$\theta w$ measured at $O_3HP$ (0.15 m$^3$ m$^{-3}$).

Figure 5: Calculated versus measured isoprene emission rates (in µgC $g_{DM}^{-1}$ h$^{-1}$) under ND (*n*=267) and AD (*n*=138) from June 2012 to June 2013, using (**a**) *G*14 (this study) and (**b**) MEGAN2.1 isoprene model (Guenther et al., 2012) with a wilting point value $\theta w$ of 0.15 m$^3$ m$^{-3}$ (measured at the $O_3HP$). The dotted line is the 1:1 line.

Figure 6: Seasonal variations of the relative contribution of the different frequencies as considered in *G*14 (0, 7, 14, 21 days before the measurement) among the regressor $x_i$ selected in *G*14, under (**a**) ND (*n*=267) and (**b**) AD (*n*=138). The frequency '0', '7', '14', '21' includes

the contribution of '$L$-1, $T$-1, SW-1, $T$0, $L$0, $T_M$-$T_m$', 'SW-7, ST-7, $P$-7', '$T$-14, SW-14, ST-14, $P$-14' and '$T$-21, SW-21, $P$-21' respectively.

Figure 7: Seasonal variations between present (2000-2010) and future (2090-2100) relative changes of SW, $P$ , ST and $T$ over the continental Mediterranean area obtained using (**a**) RCP2.6 and (**b**) RCP8.5 projections.

Figure 8: Sensitivity of the seasonal variation of isoprene emission rates calculated using $G$14 (ER$_{G14}$, in µgC g$_{DM}^{-1}$ h$^{-1}$, this study) to (**a**) $T$ and ST changes as in RCP2.6 ('T' case) and RCP8.5 ('TT' case) respectively; (**b**) SW and $P$ changes as in RCP2.6 ('P' case) and RCP8.5 ('PP' case) respectively; and (**c**) combined $T$, ST, $P$ and SW changes as in RCP2.6 ('T+P' case) and RCP8.5 ('TT+PP' case) respectively. Present and future cases were calculated for (2000-2010) and (2090-2100) respectively. Overall annual relative changes to present are framed.

Figure 9: Sensitivity of the seasonal variation of isoprene emission rates calculated using $G$14 (ER$_{G14}$, in µgC g$_{DM}^{-1}$ h$^{-1}$) to SW. Overall annual relative changes to present (2000-2010) are framed.

Table caption

Table 1: Correlations between $ER_{MEGAN}/ER$ and the soil water content (SW) cumulated over 7 to 21 days before the measurement. $ER_{MEGAN}$ and ER are isoprene emission rates calculated using MEGAN2.1 (Guenther et al., 2012) and measured (this study) respectively.

Table 2: Annual absolute and relative changes to present of SW, $P$, ST and $T$ according to RCP2.6 and RCP8.5 scenarios. Present and future cases were calculated for (2000-2010) and (2090-2100) respectively.

## Appendix 1: Calculation of isoprene emission rates $ER_{G14}$ (µgC $g_{DM}^{-1}$ $h^{-1}$) using the $G14$ algorithm

Due to the large range of ER variations, emissions were considered as logER, where:

$logER_{G14}=log[ER_{G14\ (CN)}]\times s + m$ and $s$ is the standard deviation of $logER_{G14}$ ($s$=0.8916), $m$ is the mean of $logER_{G14}$ ($m$=0.8434), $log[ER_{G14\ (CN)}]$ the central-normalised log10 of $ER_{G14}$ calculated as:

$$log[ER_{G14(CN)}]=w_0 + w_{1,k}\times \tanh(N_1) + w_{2,k}\times \tanh(N_2) + w_{3,k}\times \tanh(N_3)$$

where $N_1=w_{0,1} + \sum_{i=1}^{i=16}\sum_{j=1}^{j=16}w_{i,1}\times x_j$

$$N_2=w_{0,2} + \sum_{i=1}^{i=16}\sum_{j=1}^{j=16}w_{i,2}\times x_j$$

$$N_3=w_{0,3} + \sum_{i=1}^{i=16}\sum_{j=1}^{j=16}w_{i,3}\times x_j$$

Table A1. The optimised weights $w$ as follows:

| $w0$ | -1.29837907 | | | | |
|------|-------------|------|-------------|------|-------------|
| $w0,1$ | -0,16226148 | $w0,2$ | 2.90404784 | $w0,3$ | 0.23868843 |
| $w1,1$ | 0.07736039 | $w1,2$ | 2.18450515 | $w1,3$ | -0.1283214 |
| $w2,1$ | 0.04806346 | $w2,2$ | -0.0074737 | $w2,3$ | 0.06711214 |
| $w3,1$ | -0.32907201 | $w3,2$ | 0.31067189 | $w3,3$ | 0.14496404 |
| $w4,1$ | 0.54847219 | $w4,2$ | 0.40895098 | $w4,3$ | -1.1895104 |
| $w5,1$ | -0.03820985 | $w5,2$ | 0.27886813 | $w5,3$ | 0.35561345 |
| $w6,1$ | 0.34677986 | $w6,2$ | 0.2906721 | $w6,3$ | -2.84020867 |
| $w7,1$ | -1.44104866 | $w7,2$ | -1.23651445 | $w7,3$ | 4.30350692 |
| $w8,1$ | -0.63559865 | $w8,2$ | -0.63879809 | $w8,3$ | 3.61172683 |
| $w9,1$ | 0.81398482 | $w9,2$ | 0.85053882 | $w9,3$ | 0.46501183 |
| $w10,1$ | -2.01376339 | $w10,2$ | 1.59664603 | $w10,3$ | -0.74513053 |
| $w11,1$ | 1.61737626 | $w11,2$ | -1.68773125 | $w11,3$ | -2.29893094 |
| $w12,1$ | -0.57093409 | $w12,2$ | -0.76488022 | $w12,3$ | 1.96571085 |
| $w13,1$ | 0.78483127 | $w13,2$ | 0.9786783 | $w13,3$ | -1.88733755 |
| w14,1 | 0.05311514 | w14,2 | -0.88244467 | w14,3 | -1.90110521 |
| w15,1 | -0.47856411 | w15,2 | -0.88883049 | w15,3 | 1.35713546 |
| $w16,1$ | 0.39618491 | $w16,2$ | 0.55564983 | $w16,3$ | -0.73830992 |
| $w1,k$ | -2.22601227 | $w2,k$ | -1.64346181 | $w3,k$ | -1.32117586 |

Table A2. The selected input regressors $x_i$ as follows:

| | |
|---|---|
| $x_1$ | $L0$ |
| $x_2$ | $L$-1 |
| $x_3$ | $T0$ |
| $x_4$ | $T$-1 |
| $x_5$ | $T_M$-$T_m$ |
| $x_6$ | $T$-14 |
| $x_7$ | $T$-21 |
| $x_8$ | SW-1 |
| $x_9$ | SW-7 |
| $x_{10}$ | SW-14 |
| $x_{11}$ | SW-21 |
| $x_{12}$ | ST-7 |
| $x_{13}$ | ST-14 |
| $x_{14}$ | $P$-7 |
| $x_{15}$ | $P$-14 |
| $x_{16}$ | $P$-21 |

**Figure 1**

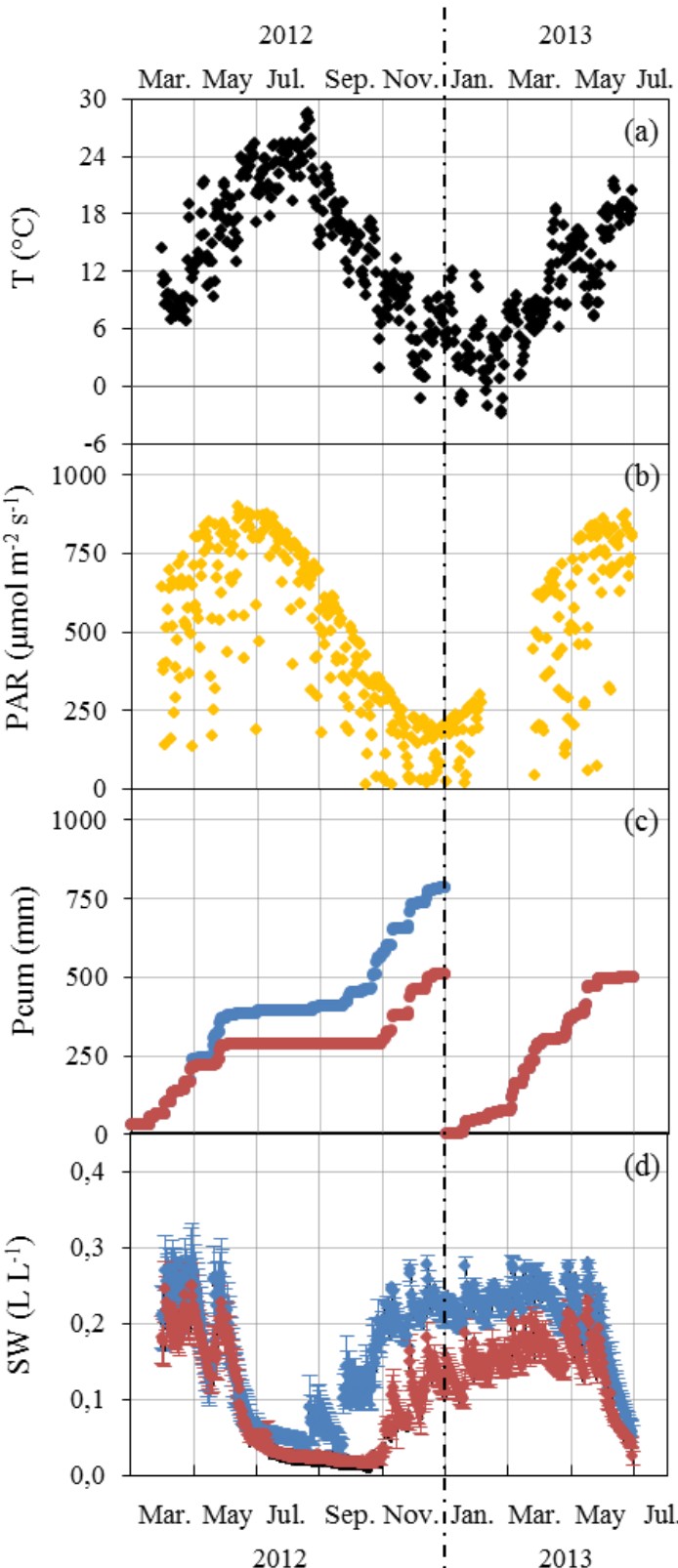

**Figure 2**

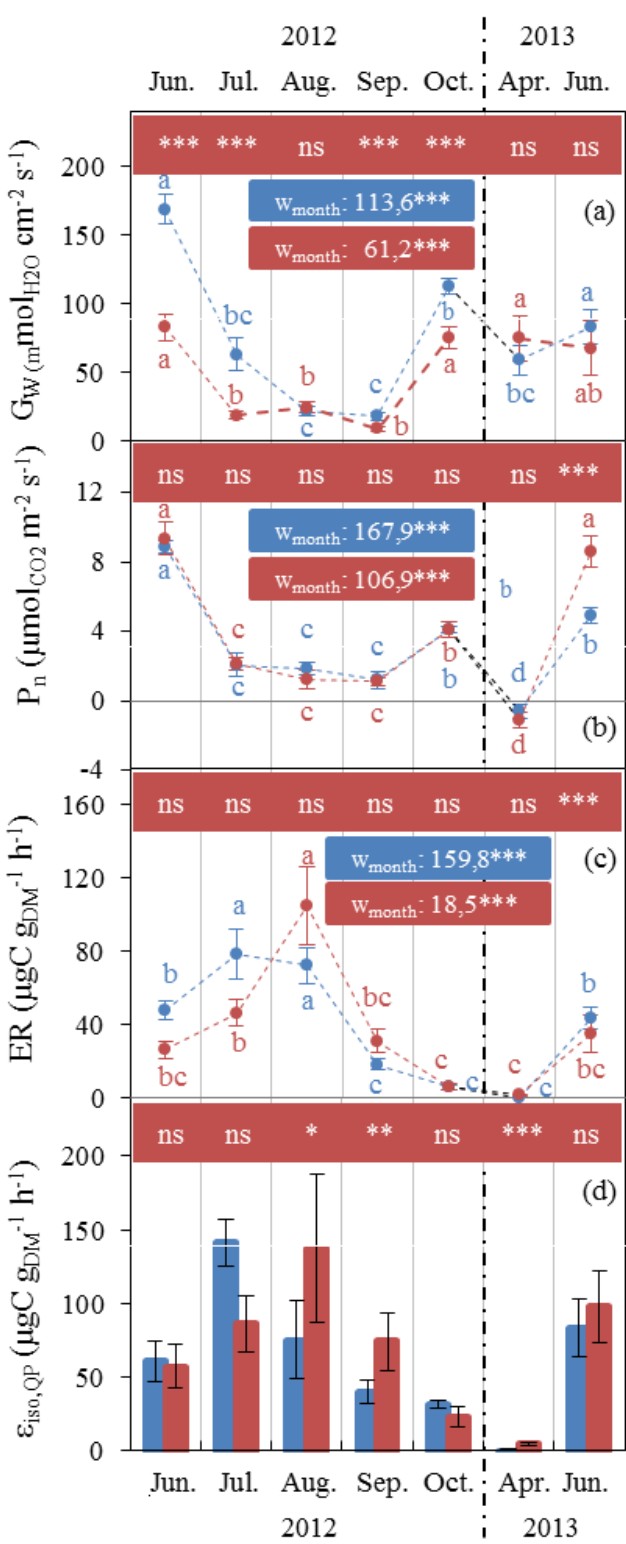

**Figure 3**

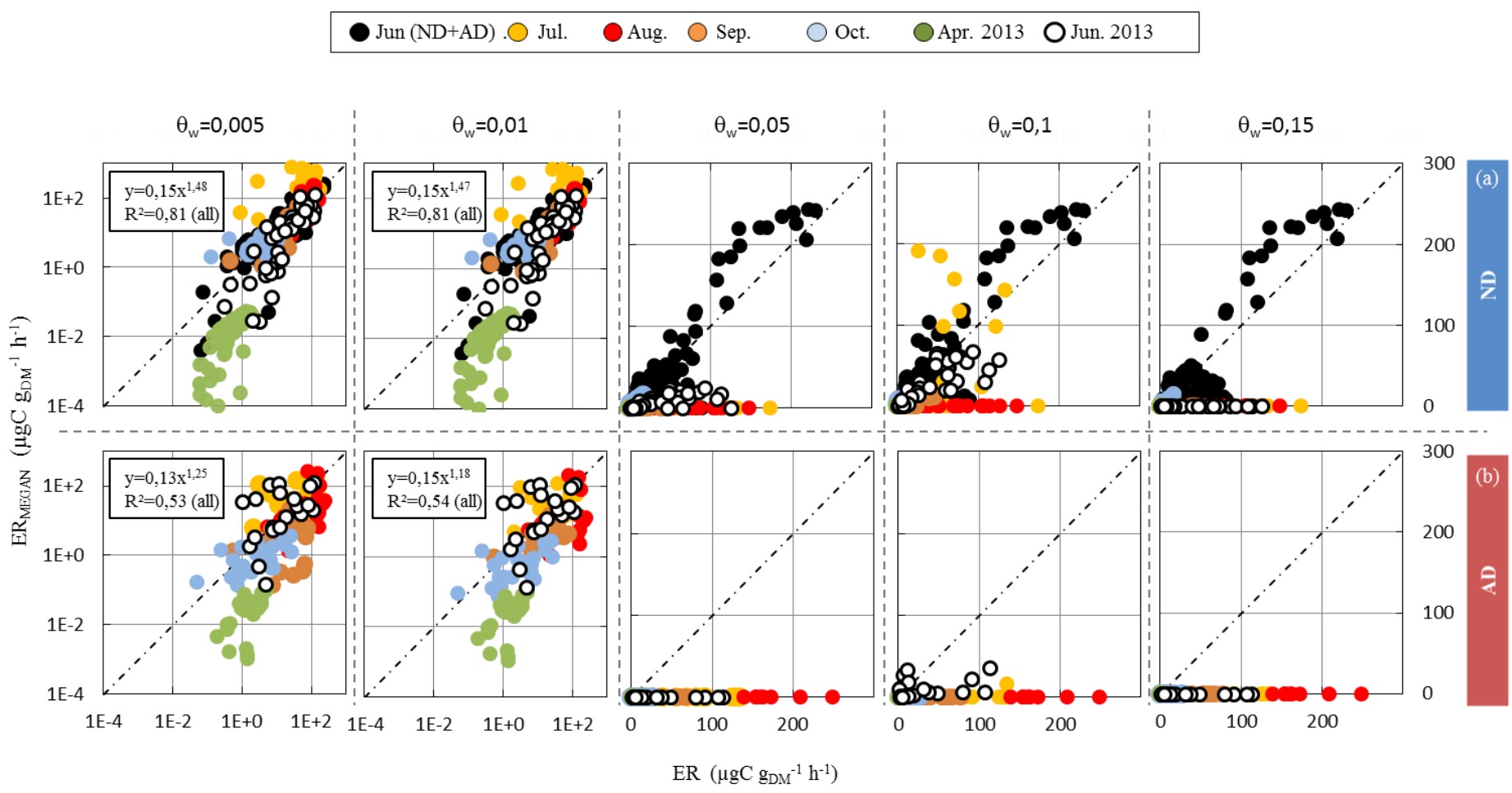

**Figure 4**

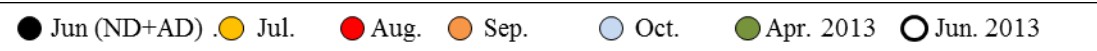

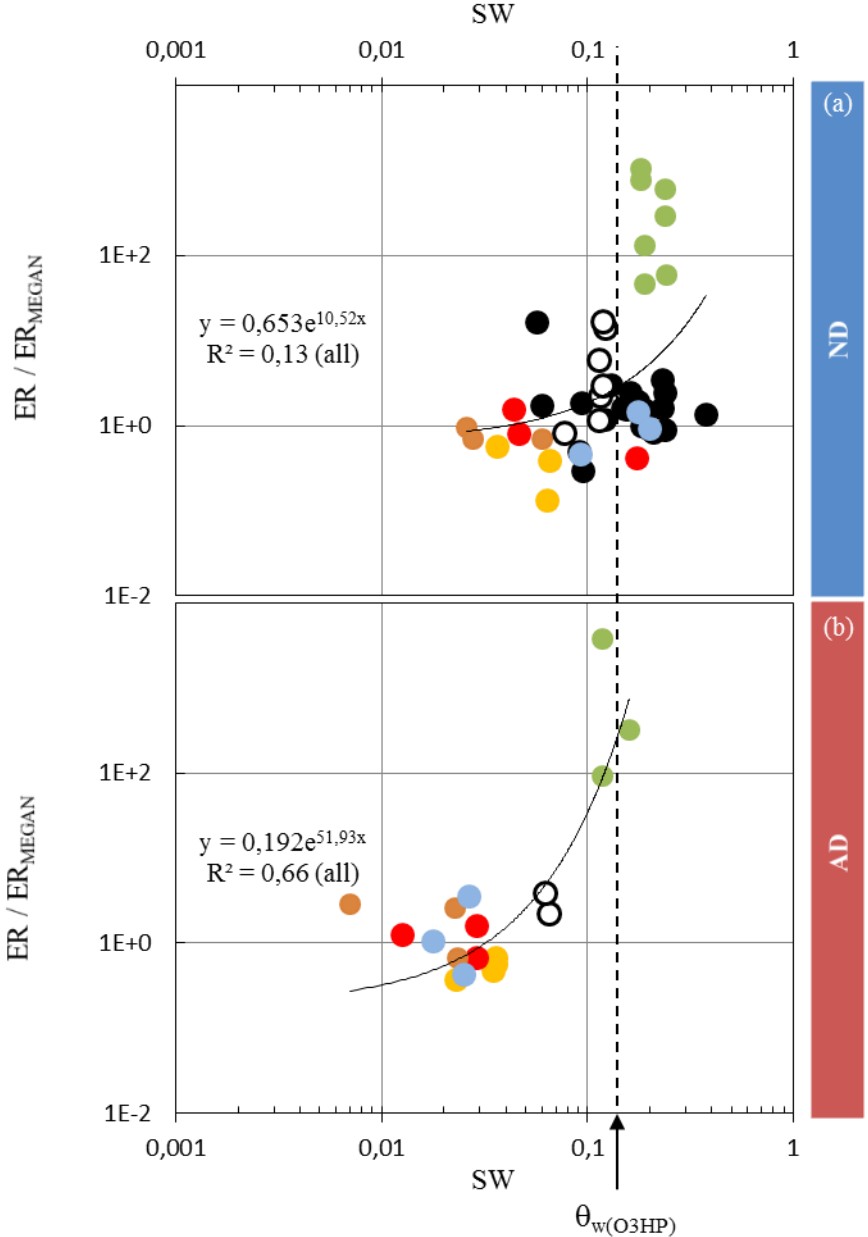

**Figure 5**

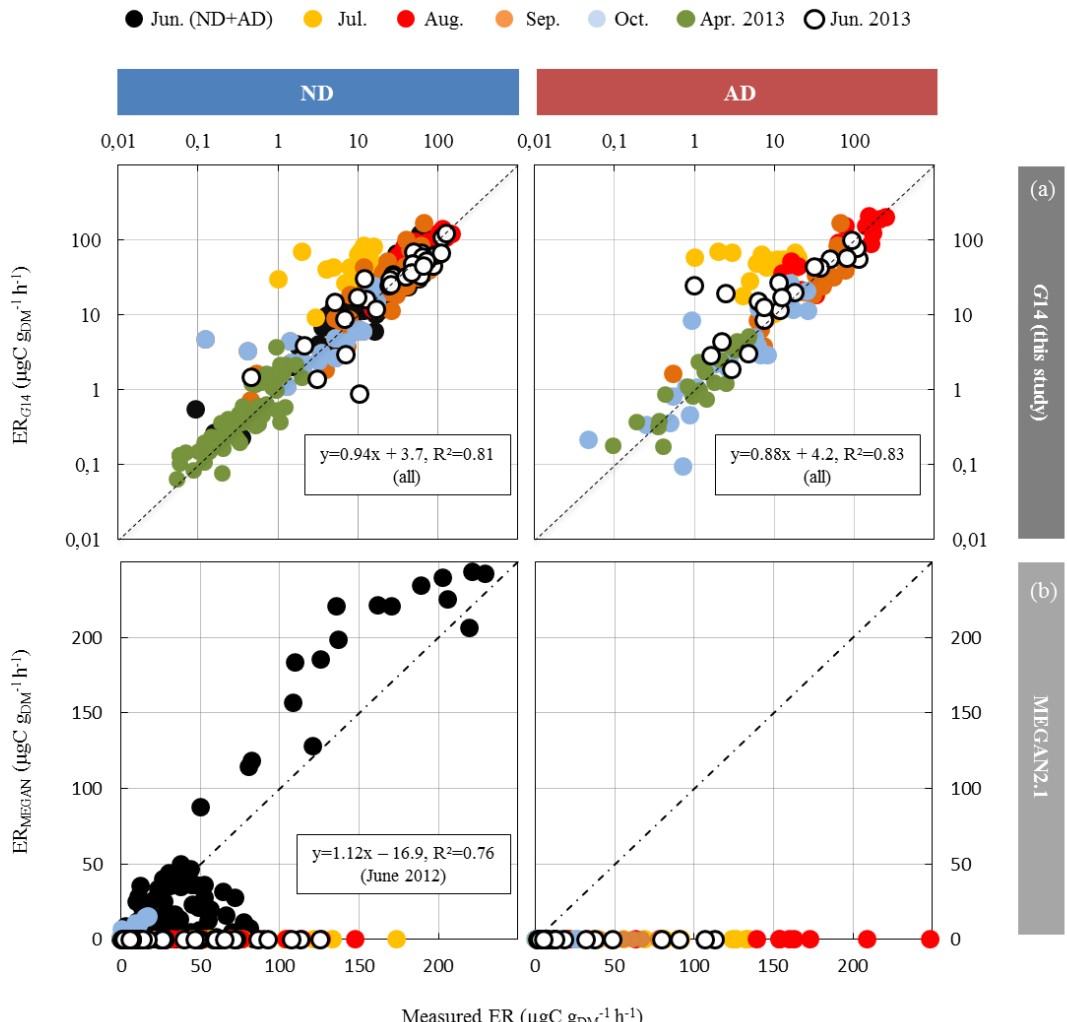

**Figure 6**

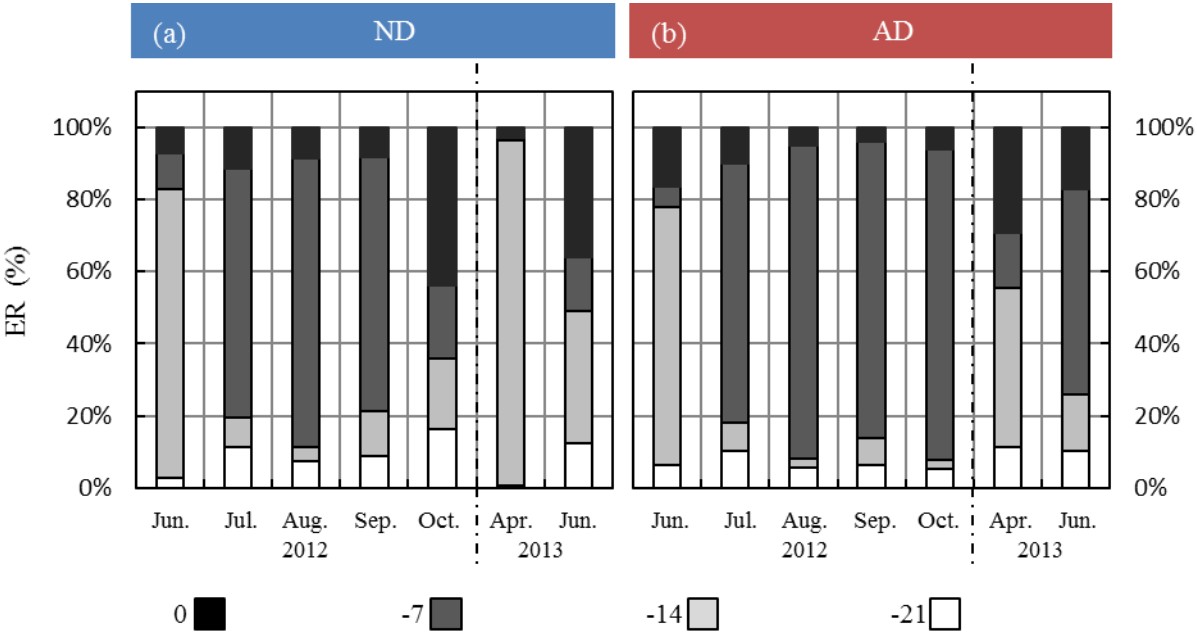

**Figure 7**

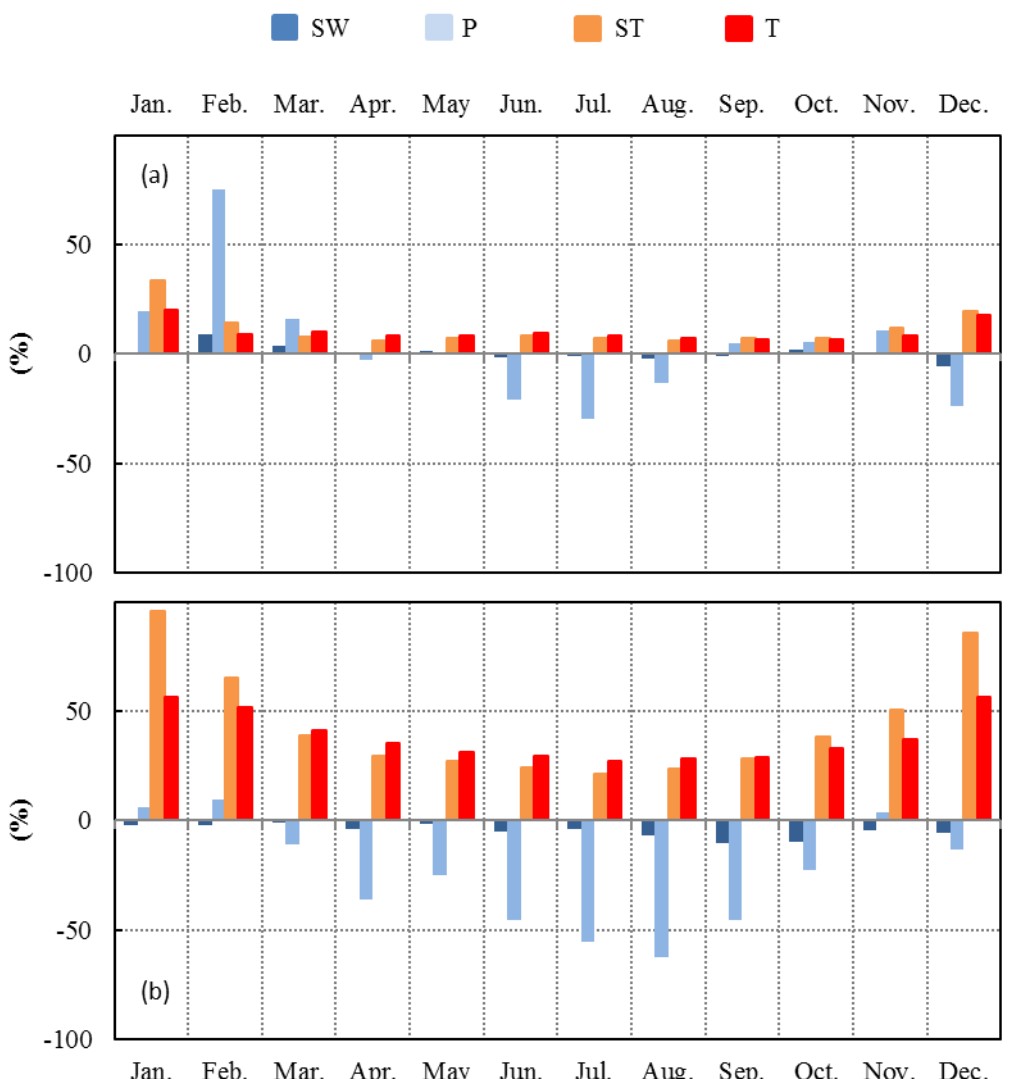

**Figure 8**

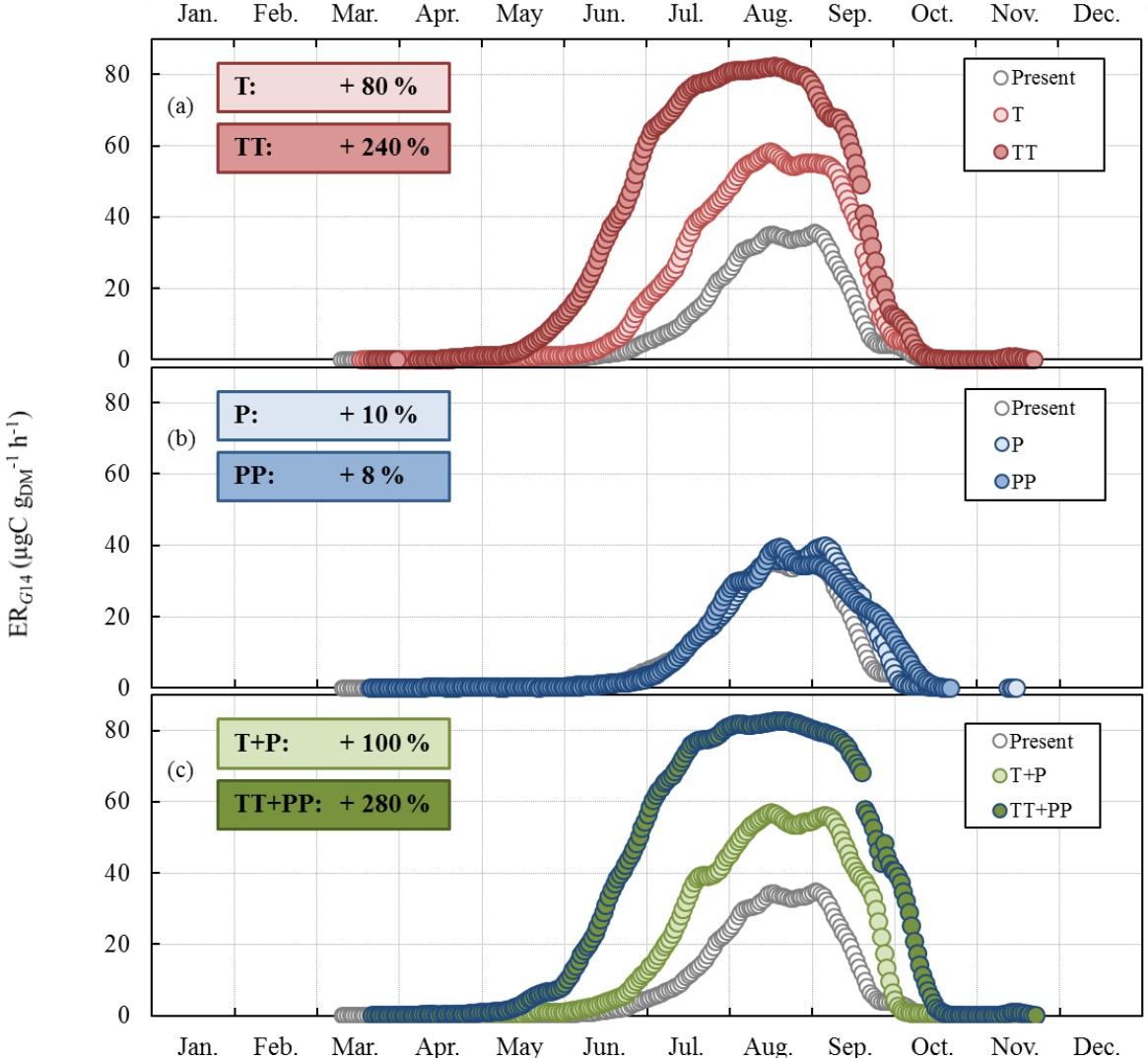

**Figure 9**

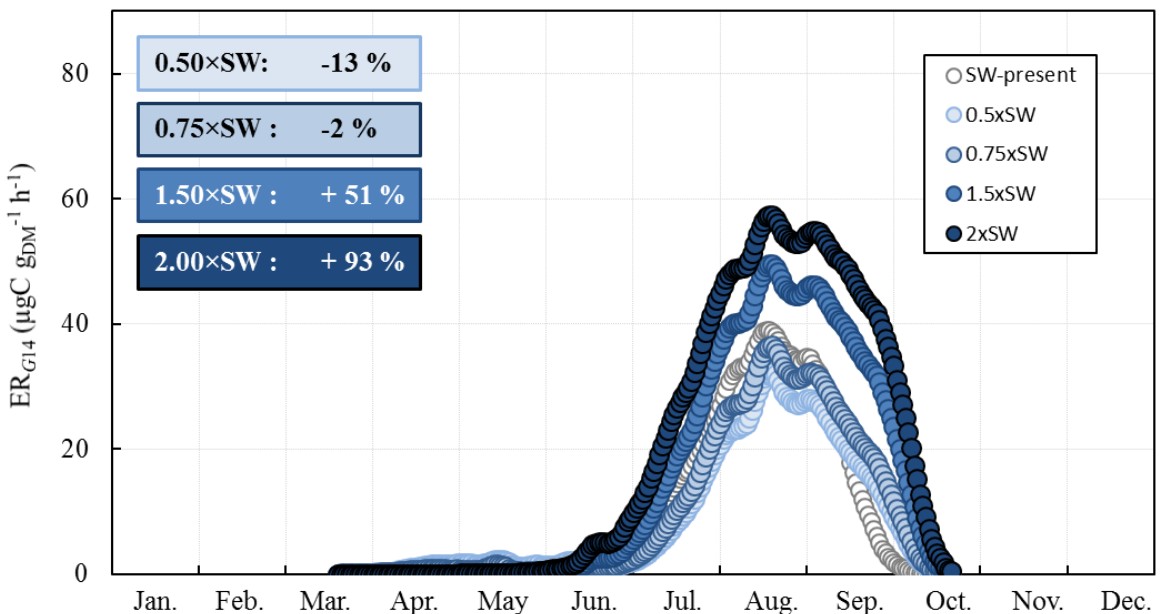

**Table 1**

| $x$ | ND | | AD | |
|---|---|---|---|---|
| | ER/ER$_{MEGAN}$=f($x$) | R² value | ER/ER$_{MEGAN}$=f($x$) | R² value |
| SW | $0.653e^{10.5x}$ | 0.13 | $0.192e^{51.1x}$ | 0.66 |
| SW-7 | $0.715e^{1.30x}$ | 0.13 | $0.239e^{6.30x}$ | 0.55 |
| SW-14 | $0.763e^{0.57x}$ | 0.11 | $0.279e^{2.74x}$ | 0.48 |
| SW-21 | $0.523e^{0.46x}$ | 0.14 | $0.365e^{1.47x}$ | 0.38 |

**Table 2**

| | $\Delta$SW (m$^3$ m$^{-3}$) | $\Delta Pcum$ (mm) | $\Delta$ST (° C) | $\Delta T$ | $\Delta$SW/SW | $\Delta Pcum$/$Pcum$ (%) | $\Delta$ST/ST | $\Delta T$/$T$ |
|---|---|---|---|---|---|---|---|---|
| RCP2.6 | + 0.004 | + 30 | + 1.4 | + 1.4 | + 0.5 | +5 | + 8.4 | + 9.1 |
| RCP8.5 | - 0.007 | + 30 | + 5.3 | + 5.3 | - 5.0 | -24 | + 32 | + 34 |