# Peer review of "Seasonal variations of *Q. pubescens* isoprene emissions from an *in natura* forest under drought stress and sensitivity to future climate change in the Mediterranean area."

_Biogeosciences, 2017_

## Referee Comment (RC1) · Anonymous Referee #1 · 9 Mar 2017

Abstract

Lin 1: Please, change physiology by @gas exchange@.

Lin 20-23: In order to reflect the seasonality in this text, please indicate the lowest values in september and october...

Lin. 20: is not the lowest in April 2013

Lin. 22: Isoprene emission factor and Isoprene emission rates.

Introduction

[Figure]

No comments

Materials and Methods

Pg. 5 Lin. 10 What volume was sampled for each cartridge?

Pg. 8 Lin. 7: HTTP Error 403.14 – Forbidden The Web server is configured to not list the contents of this directory.

Results

Pag 9 Lin. 31: in Fig. 1b shows in May

Pag. 10 Lin. 20: Consider the deepest roots to take water. . ..

Lin. 26: Consider the senescence of leaves.

Discussion

It would be well that in the discussion the authors should take into account that the trees studied can obtain water through the deep roots. Have you determined in any way to get water from the subsoil by Q. pubescens?

On the other hand there are more studies that expose the relationship between moderate water stress and the increase of emissions of BVOCs.

Studies that demonstrate the effect of light as an activator of isoprene synthase may also be considered. Are the authors supposing that the activation of isoprene synthase can be due to factors other than light? Could they justify it?

Pag. 13 Lin. 15: put Q. pubescens in italics.

Pag. 15 Lin. 12: 2014 is missing the first parenthesis.

Pag. 16, Lin. 9-16: I repeat the importance of taking into account the water supply through the subsoil.

Figure caption

If possible, keep the colors of the legends in all graphs.

Pag. 23 Lin. 3: Instead of July 2013 is June 2013.

Pag. 23 Lin. 9: Instead of June 2012 to 2013 write June 2012 to June 2013.

Pag. 23 Lin. 15: Instead of "among months" would not it be better to say "among water stress treatments"?

Pag. 23 Lin. 18: Instead of June 2012 to 2013 write June 2012 to June 2013.

Fig. 1: It would be very good if the colors differ according to the water treatment.

Reissue the legend of the axis and removing points and putting when necessary. In addition, a horizontal line would help visually to differentiate the years.

Fig. 2: In the legend of the y-axis of the graphs c and d put "isoprene".

Fig. 3. Could they indicate the regression lines and R2? And also indicate to which corresponds each color.

Fig. 4: In the legend of the y-axis it would be better to put: Isoprene ER (%). In addition, at the bottom of the Figure could mark the years of sampling.

Fig. 5: In the legend of the y-axis of the graph (a) put (%) In the legend of the y-axis of the graph (b) put "Isoprene" On the axes x enter the legends of the abbreviated months name.

Pag. 24 Lin. 5: Instead of October is November.

Pag. 24 Lin. 8: Indicate what corresponds to each abbreviation (inst and cum).

---

## Referee Comment (RC2) · Anonymous Referee #2 · 22 Mar 2017

General comment: The subject of this paper is important to address. Water stress on plant BVOC emissions is generally lacking or under-represented in models. Field observations in the regions where plants suffer from drought and have high BVOC emission are important for understanding processes beneath. The paper has strengths, but also needs major improvements before considering publication. I am a modeller, so my main focus will be on the modelling part in this paper. The collected data have clearly shown the impacts of AD treatment on isoprene emission through the season. However, the application of MEGAN in the paper is less satisfying. For the key parameters to model isoprene emission in this region, instead of using site-observed data,

the authors picked values from literature. And then there is long discussion about how unsuitability of MEGAN for this site. I would strongly suggest the authors to use site-specific data before evaluating the MEGAN model. And then, the future runs with the trained G14 model are interesting, but the discussion has put too much focus on the number coming from the runs, instead of really discuss potential uncertainties from the applied G14 model as well as relating these numbers to other predictions, like process-based model to address limitations of using ANN method. At last, this paper has no conclusion. The key messages to readers are not clear at this moment.

Abstract: It clearly described what results this paper has gotten, however, the conclusion is currently missing. There are quite many abbreviations in abstract as well as in the main paper, which makes the reading less smooth. Furthermore, the readers could also like to have a few lines at the beginning of the abstract to know why we need this type of study.

Method: First, There is no description about MEGAN model at all in the method section. Second, the purpose of running ORCHIDEE model is to get the predicted SW and ST for using in G14, right? How this modelled data have been used is not described. Then, for the AD treatment, is the deployable roof blocking all precipitation during certain period? If not, the authors need to clarify how this AD treatment has been done.

Results: When applying MEGAN model for this study site, why not adjust the emission capacity and wilting point based on data form this study site? There is large section in the discussion about the unsuitability of wilting point used in MEGAN for this study site. But the model is not designed for this specific site, I think it is logical to use the site-specific data before discuss potential uncertainties in MEGAN.

Discussion: The authors have put a lot of text on discussing the simulated ER in future scenarios, which I don't think it is necessary. The trained G14 model, with only influences from environmental factors, has no considerations of C source for isoprene synthesis, vegetation dynamics as well as adaption. The training period for G14 is too

short to be able to see the full picture of potential future dynamics of ER in this region. I would rather see the runs on the future scenarios as sensitivity testing, not prediction. There is no conclusion in the main paper.

Detailed comments:

Change "gaz" to "gas".

P2, L7, spell out species full name when it is first time mention.

P3, L5, need references for empirical-based model.

P3, L31, take away comma before and after "both".

P3, L32, use abbreviation for species name after the 1st time.

P4, L3-8, the aims seem to list what this manuscript is going to do, not really to any level of scientific understandings. I would suggest the authors to promote the real scientific questions this manuscript is aiming to answer.

P4, L26, "during one week, once a month" suggest to change to "one week per month"

P4, l27, "except from" suggest to change to "except for the period from"

P5, L22, have the biases between the Forcalquier station precipitation data and the site data been adjusted before using?

P6, L18, what is "AF"? Too many abbreviations in this paper! And the abbreviations are not consistent sometimes, e.g., both SW and Sw have appeared in the paper.

P6, L23, wrong order of Cout and Cin.

P8, L1, how does Krusal-Wallis test use for detecting seasonal variation?? Please clarify it.

P9, L31, what do you mean "the amplification of the ND" at the beginning of this sentence. In the same sentence, how can we see the ND changes reach maximum in July

and continued till Nov? Please clarify it.

P10, L2-5, why there is an increase of Pcum in the AD plots after Sep, 2012? so the rain is not totally blocked?

P10, l11, On the P5, L17-19, it mentioned that the same PAR was used for AD and ND plots. Why there is difference existed? Please clarify it.

P10, L22-23, suggest to delete it, it is for discussion and has been mentioned as well.

P11, L2, what "P = 3.9" mean? statistic p-value?

P11, L5-9, "the general high variability observed . . ." it is hard to see high variability for the AD-ND site in April, what do you mean or refer to here?

P11, L13-14, why not adjust the emission capacity values and wilting points based on the site-specific data?

P11, L18, not accurate to say "correctly assessed".

P12, L6-7, "were more sensitive to lower frequencies. . ." is not correct. From the figure, for this period April and June, ER is still more sensitive to the 0-7 days changes, but just with relatively more contributions from 14-21 days, compared to the situation in July. Right?

P12, L9, I could suggest the authors to convert the contribution from each factor into ratio (%), instead of absolute values in ER. After conversion to ratio, the authors could compare the changes of explanation ability of each factor through the season.

P12, L30, wrong order of the numbers for corresponding RCPs

P12, L31, should be dERG14

P13, L17, "during all summer" change to "during the summer"

P14, L5, both "$\mu$gC gDM-1 h-1" and "$\mu$g gDM-1 h-1" have appeared in the paper. Please correct them and be consistent.

P14, L28-31, "other additional parameters. . ." : I would suggest the authors to change the conclusion made here, before adjusting the MEGAN model to the local conditions.

P15, L21-24, the sentence starting with "Note that MEGAN performed better on our experimental data . . .", since the authors already know that the wilting point used in the MEGAN is not suitable for this site, the impacts from SW is far-off at this moment. As mentioned above, it is better to adjust the parameters used in MEGAN first and then discuss potential uncertainties existing in the model about stress enhancement on isoprene emissions.

P16, L2-3, change " Ìťİř" to the standard used in BG. And the degree day is not the first time mentioning, put the abbreviation at the place where it shows up the first time. P16, L12, ". . . representation of soil moisture in . . . is currently poor". I don't think it is relevant here to mention the quality of soil moisture estimation at large scale. It may be not the case for the study site. Also, there is no soil moisture validation for models in this paper. It may be more relevant to discuss about the response function of soil moisture (the one multiplying with emission capacity).

Section 4.3. In general, it is unnecessary long section, with many parts of discussion is describing results (e.g.,

P17, L4-6, P17, L13-16). Since the trained model G14 has so many uncertainties in terms of predicting future emissions, like, short training period, the ignorance of C source for isoprene synthesis, spending long text to describe how much changes predicted from this model is not necessary. As mentioned at the beginning, I would take this exercise as a sensitivity testing, not prediction. It could be more relevant to compare the testing result from this paper to other literature for the same or similar area and then further suggest what could be further improved in this ANN method.

Figures:

Figure 1: it is difficult to link data to the xtick/xlabels.

Figure 2: what the letter "c" mean?

Figure 3: Why not do the same log-transform for the comparison between the modelled and measured ER? There are many months' data which cannot be shown on the Fig. 3a. Suggest to change them.

Figure 5: check if all abbreviations have been explained in the caption.

---

## Referee Comment (RC3) · Anonymous Referee #3 · 31 Mar 2017

This manuscript presents the findings of an analysis of isoprene emissions from oak trees measured during an extended field campaign in a Mediterranean forest. Trees were sampled from two plots, one covered to intercept rainfall and thus subjected to artificial drought conditions and the second left open and therefore representative of normal water conditions. In this region, even the trees in the control plot experience drought during the summer months. Artificial neural network analysis was used to determine the meteorological and physiological parameters that most strongly influence isoprene emissions throughout the course of the growing season. The network was optimised and tuned for the site, and its skill assessed using reserved measurements

taken from the same trees and sampling period.

With dry regions projected to become drier and experience more prolonged and intense periods of drought in the future, it is important to understand the response of native vegetation to such conditions. If we are to assess the potential impact of changes in vegetated ecosystems on climate, air quality and the Earth system in its entirety, it is equally important to improve the skill of current models in capturing the processes and interactions under present-day conditions. A study that combines observational data with model evaluation and development would therefore seem to represent an ideal approach. However, the study presented here, appears to my mind to have major weaknesses in design and implementation that preclude publication in its current form. Considerable work would be required to sufficiently overcome these limitations.

I also have concerns regarding the presentation of the work, results and conclusions in terms of both language and style. Given the degree of re-working and re-writing required I will limit my comments to suggesting that the authors would do well to have the manuscript proof-read and edited by a native English speaker as there are times when the grammar and choice of word make it hard to follow. In addition, the way in which the results are presented is often non-sequential, jumping from one variable to another rather abruptly and without clear logic again making it hard to follow. There are also times where the authors appear to contradict, or overlook, points they have made earlier in the article but it is hard to judge whether this is a consequence of my misunderstanding them.

My main concerns are outlined below:

Background and citations The authors appear unaware of a large body of work previously conducted into the effects of drought on photosynthesis, cellular processes and emissions of volatile organic compounds such as isoprene. The studies they cite are narrow in focus and scope. While this is of course appropriate when reporting specific emissions factors or observations from similar ecosystems, I would expect to see a

far wider discussion and consideration of previous modelling studies particularly those that apply "MEGAN" to simulate isoprene emissions. For example, the authors do not appear to consider the excellent evaluation of the skill of these empirical algorithms in capturing observed seasonal variability of isoprene emissions (Müller et al., 2008) which also investigated whether this was attributable to soil moisture and the implications for estimates of global emissions; nor the assessment of the treatment of wilting in the MEGAN algorithms (Sinderalova et al., 2014).

Measurements 1. It appears that measurements of PAR and temperature were almost exclusively made above the top of the canopy. It is not clear to me that this would be representative of the conditions within the forest canopy, i.e. those experienced by the majority of the foliage. While the authors do state that the enclosed branches were toward the top of the canopy net primary productivity and canopy stomatal conductance reflect conditions experienced by all leaves. Furthermore, photosynthesis, respiration and isoprene emissions are all strongly affected by actual received radiation and leaf temperature yet the authors confine their analysis and model development to air temperature and top of canopy radiation. How have they accounted for shading within the canopy, which will affect leaf temperature as well as available light? Or the occurrence of sunflecks?

2. How exactly are induced drought conditions achieved? How is the roof operated? The overall effect of the deployment of the roof might be the roughly 30% reduction in annual precipitation expected in the area under climate change but how is this reduction distributed? Evenly, i.e. a 30% reduction every day or every month? It is not just total rainfall that affects vegetation physiology and phenology, it is also the temporal pattern. The overall drying of the region is expected to result in more severe prolonged droughts interspersed with periods of increased intense rainfall. Is this reflected in the artificial drought conditions produced at the site?

3. Too little detail of the sampling strategy is given. A full list of the dates of isoprene measurements is required. It would seem to be far from the one week per month from

June 2012 to June 2013 that is stated. It is also unclear whether cartridge samples continued to be made when the PTR was deployed, and if so whether they were also taken at multiple heights. If the techniques were deployed in parallel were the data compared for consistency? How were the data from lower in the canopy included in the analysis, given the issues I have raised in point 1?

4. More details should be given of the use of the data from COOPERATE. Specifically which parameters were selected and on what basis? How exactly were precipitation data from the nearby site used to gap fill the COOPERATE data? Precipitation is highly heterogeneous in both time and space, particularly in mountainous regions such as the Haute Provence. Were data compared from times when both datasets were available?

Statistics 1. "MEGAN" - The algorithms referred to as MEGAN have evolved over time from the initial parameterisations based on leaf-level emissions measured under controlled laboratory experiments presented in 1991 and 1993 by Guenther et al. Over time these have been extended and adapted to represent canopy-scale emissions. The authors appear unaware of the major differences implied by the changes, and are inconsistent in the set of algorithms they choose to apply. The authors begin by back-calculating isoprene emission factors for each month under standard conditions – using the leaf-level parameterisations which is entirely appropriate. (Although, as the authors specifically refer to "CL" and "CT" on numerous occasions I feel that they need to present the equations they are using here in the text rather than referring the reader to the original papers.) It should be noted that other studies have found wide variation in "standard" emission factors between different leaves on the same tree, let alone different trees yet the authors have sampled only a single uppermost branch from each tree as noted above. Later, the authors switch to using the MEGAN algorithms from 2006. However, they appear unaware that firstly these algorithms use different emission factors from the leaf-level ones calculated from the 1991/3 parameterisations as they account for canopy architecture (shade and sun fractions, leaf angle distribution, vertical distribution of foliage) and in-canopy losses. To compare emission rates estimated with MEGAN against the measured single branch emissions does not appear a "fair" assessment (given the different approach taken to assess the estimates from the "G14" algorithms – see below) MEGAN (both v2.0 and 2.1) include a dependence on "historical" PAR and temperature, the average conditions over the previous 24 hours and 10 days. This is similar to the findings of the artificial neural network analysis here: that generally emissions are most sensitive to fluctuations in driving data occurring over a period of 7-14 days, but that at times very short time-scale processes dominated the effect. The authors have not acknowledged this feature of the MEGAN algorithms and in fact appear to state that there is no "memory" in MEGAN.

2. Using an artificial neural network approach to deduce an algorithm for isoprene emissions essentially produces a best-fit parameterisation that is tightly tuned to a single site and single time period. One would expect such a parameterisation to show skill for those specific conditions. Given the sparseness of the data used for this tuning the robustness of using the resulting model to estimate emissions under different conditions is not self-evident. At the very least, data from a much longer time period (and ideally more than one location) is required to give confidence of the capability of the model to capture e.g. emissions in 2100 under RCP8.5.

3. The artificial neural network approach is limited in that while it highlights the variables to which, in this case, isoprene emissions are most sensitive, it does not provide insight into the fundamental processes that influence the emissions most strongly. Therefore, it is hard to use the knowledge gained to improve existing understanding or modelling.

Modelling Which brings me to what is probably my chief concern, the inconsistency in the two approaches to modelling emissions under future conditions. The "G14" algorithm, the optimised statistical parameterisation derived from the artificial neural network analysis, is tuned specifically for this site, this time period and these environmental conditions. and the authors demonstrate that it performs well for this site, this time period and these environmental conditions. By contrast, and in spite of the fact they have site-specific data available, they apply the MEGAN algorithms in the default

form, i.e. with generic emissions factors and wilting point threshold. Why? It is not clear what we gain from this assessment. Müller et al. 2008 and Potosnak et al., 2014 have already demonstrated that when applied in this way, MEGAN does not capture observed seasonality or response to water stress, and Sinderalova et al., 2012 show that changes in soil wilting point threshold can alter estimated global isoprene emissions by up to 50% annually. It would have been far more valuable and far more consistent had the authors taken the same approach as they did with G14 that is to "tune" the model to the site. How does MEGAN perform if the site-specific monthly varying isoprene emission factors are used? Or if the authors experiment with different wilting point thresholds? Given that G14 is not to my mind robust enough to apply under different conditions we are still left with a need for a set of algorithms that can be applied globally and over extended time periods, such as MEGAN. However, I acknowledge that we do need to improve the skill of these global algorithms to replicate observations, particularly under periods of environmental stress that could be anticipated to occur more frequently in the future. The authors spend too much time presenting and discussing the results of future simulations using the G14 algorithms in light of the weakness of applying such a tightly tuned model under future conditions. Better still if the authors were able to leverage the G14 artificial neural network analysis to develop a process-base model of the effect of soil moisture on isoprene emissions, or contribute to efforts to improve such a model, e.g. Gröte et al., 2010.

ORCHIDEE – It's not clear whether the authors used ORCHIDEE to estimate soil moisture content and soil temperature for the present-day as well as under the RCP scenarios, given that these variables were measured on-site (in fact, around the base of each sampled tree) throughout the growing period. What met data were used to drive ORCHIDEE future projections? What downscaling techniques were applied to back out high-resolution precipitation and other meteorological variables for the location of the measurement site?

My recommendation to the authors would be to use the site-specific data to deduce

monthly emissions factors and wilting point values to "optimise" the performance of the current (i.e. Guenther et al., 2012) version of MEGAN for this site, and to evaluate the optimised model. But to concentrate their efforts on using the artificial neural network analysis to attempt to gain insight into the fundamental processes that give rise to the observed changes in isoprene emissions. At present they are only able to draw on hypotheses from previous studies to try to explain the variations but cannot support or repudiate the hypotheses. I would suggest they use the same approach to explore the drivers of other physiological parameters: stomatal conductance, sap flow, transpiration, water, carbon and energy fluxes to determine whether the responses of any reflect the same drivers as isoprene emissions (including frequency / speed of response).

---

## Author Comment (AC1) · 12 Jul 2017

Referee #1

General comments

None

Detailed comments

We do thank Referee #1 for his/her comments.

[Figure]

We do agree with all comments / changes suggested and will do the appropriate changes, but:

- Abstract:

Lin 20-23: the lowest ER values in October and April were 6 and <2 $\mu$g gDW-1 h-1 respectively

Lin 20: no, the lowest Gw values were observed between July and September (<20 moleH2O m-2 s-1) not in April (figure 2b)

Line 22: we do not understand this point; emission rates ER (measured values) are different from emission factors (ER normalized to temperature and PAR)

- Materials and Methods:

Pg 5 Lin 10: sampling volume varies between 0.45 and 0.9 L depending on the season and, thus, the expected emission intensity.

Pg 8 Lin 7: indeed Netral site web is no more available and we do not have another web link.

- Results & Discussion:

Pg 10 Lin 20: Electric resistivity tomography measurements have shown the heterogeneity of the karstic substrate organized as soil pockets developed between limestone rocks. Water and nutrient pools and dynamics probably differ greatly between the shallow upper soil layers and the soil pockets developed between limestone rocks. However, the soil trenches carried out in the site have shown that a calcareous slab often developed at a depth of 10-20 cm and that the roots of the oaks were rather distributed in this superficial humiferous horizon, and that only a few large roots cross this slab.

Pg 10, Lin 26: Consider the senescence of leaves: although senescence had just begun during this sampling period, but we did check that the enclosed branches were not senescent during our measurements.

- Figure caption:

The same colors of the legends in all graphs: we used blue and red colors for ND and AD respectively in all graphs; we are not quite sure what the referee wants.

Fig 1: in Fig. 1 (and other figures as well) the colors do differ from ND (blue) and AD (red) treatment. However, PAR and T values being the same for both plots, this color code was not used. In addition, a horizontal line would help visually to differentiate the years. Does the referee mean a vertical line?

Fig 3: regression lines and $R^2$ (already given for G14) will be given also for MEGAN, but, in the ND plot, only for June and October 2012 (0.25x+7.7, $R^2$=0.32 and 1.4x+0.45, $R^2$=0.82 respectively), and April and June 2013 (1.1x+0.09, $R^2$=0.65 and 0.09x+3.6, $R^2$=0.41 respectively); in the AD plot only for April 2013 (0.24x-0.15, $R^2$=0.79 (see also reply to Referee#2, Fig3). The month corresponding color is already given between the (a) and (b) graphs.

Fig. 5: % is already mentioned; the legends of the x axis is already given (on the top of the figure)

Referee #2

We do thank Referee #2 for her/his careful reading and useful comments/suggestions.

General comments

- Use of literature values instead of site-observed data: G14 was tuned using O3HP data, not literature data; this point being also mentioned by Referee #3, it will be more clearly stated in the revised manuscript.

- No discussion of potential uncertainties from G14 and ANN in general: due to the overall words limitation we did not develop this important aspect of the work; a discussion will thus be added especially in the section 2.6 of the revised manuscript. Among the other available statistical methods, ANNs present the advantage of being the most

parsimonious (e.g. giving the smallest error for a same number of descriptors; see for instance Dreyfus et al., 2002). Moreover, ANN approach, as the other non-linear regression methods, is not, or not very, sensitive to regressors' co-linearity (Bishop, 1995; Dreyfus et al., 2002). One of the ANNs limitations is that they can be used only for interpolation, not extrapolation exercises. For this reason, our future RCP projections we made using only xi values that did fit into the range of variation of xi obtained during the training phase; in total 21% of data were thus rejected.

ANN optimization during the training phase was based on the reduction of the root mean square error (RMSE) between calculated and measured values. Our final optimized RMSE (validation data) was 8.5 $\mu$g gDW-1 h-1 for our ER values ranging between 0.06 and 113 $\mu$g gDW-1 h-1, and represents 35% of the mean (22.7 $\mu$g gDW-1 h-1).

- No comparison with process based model: as mentioned p3 in the first paragraph, p3, process based model require a complex set of data to be ran; such a dataset is not available for this study. This is the reason why an 'empirical' model (MEGAN) was solely tested.

- No conclusion and no clear key messages: a separate conclusion will be added in order to emphasize on: (i) the effect of a natural and amplified water stress on isoprene emissions from Q. pubescens (an increase of the emission factor), (ii) the difficulty of current empirical emission models to describe the isoprene emissions under stress conditions, (iii) the sensitivity of isoprene emission under water stress to soil water content, thermic stress and high frequency changes, on the site investigated.

- Abstract:

Too many abbreviations: all abbreviations used are explained in the abstract (as well as in the main paper) ; they were used in order to give the maximum information in the limited number of characters of the abstract.

Few lines at the beginning to know why we need this study: we will add a sentence explaining that 'although strong isoprene emitters such as Mediterranean oaks are strongly affected by increasing drought and temperature, very little in situ information is available on how isoprene emissions will be impacted and how well the actual isoprene emission models can assess such impacts'.

- Method:

No description of MEGAN : indeed, MEGAN is generally not described when referred to, since it is widely used and known in the 'BVOC community'. A brief description with some important equations (see also Referee #3 comment) and how we 'tuned' MEGAN in our study will be added in an new appendix.

No description on how ORCHIDEE data were used: predicted SW and ST were indeed assessed by running ORCHIDEE; as explained section 3.4 line 12, they were then used in the G14 algorithm in order to project/estimate isoprene emission under two future climatic scenarios. better describe how AD roof was operated: this point is also mentioned by Referee #3; in the revised manuscript, we will explained that:

'A rainout-shelter above 300 $m^2$ of canopy dynamically excludes precipitations (rain, snow and hail) since May 2012 by deploying automated shutters; the intercepted water was collected in a belowground reservoir which could be fed into an 'irrigated' control plot during times of exceptional drought periods (under construction). In the present study, the rainout shelter was deployed during rain events only from May to October 2012 in order to exclude 32 % precipitation in the rain exclusion plot. Almost all the rains in late spring and summer were thus intercepted. Using ombrothermic diagram (P<2T, with P=monthly precipitation in mm, and T=monthly air temperature in °C) we assessed that the summer 2012 drought period reaches 4.5 months in the AD plot, compared with 3 months in the ND plot. This percentage matches IPCC projections by the end of the year and fits the precipitation reduction at O3HP during the driest years from 1967 to 2000, compared to the average precipitation of this period (see figure I

below).

Fig. I: Averaged cumulated precipitation at the O3HP between 1967 and 2000 (blue line with upper and lower standard deviation), and during the driest years of the same period (red line with upper and lower standard deviation). Cumulated precipitation in 2012 in the rain exclusion plot is represented by the green line and in the exclusion plot by the purple line.

- Results:

Use of on-site data to apply MEGAN rather than 'literature' data: this point being also raised by Referee#3 , a detailed – and common - answer is given in our reply to Referee#3 ('Main concerns', last paragraph)

- Discussion:

Long text on future ER although no consideration on C source + training period for G14 is too short; C source was indeed not directly considered since the hypothesis we made in applying a neuronal approach to statistically analyze our data was to consider 'simple' integrative environmental parameters; C source, nor vegetation adaptation, were thus not considered, but in the end we hypothesized that they are, more or less indirectly, driven anyway by PAR, T, SW and ST fluctuations over different range of frequencies. As vegetation adaptation is concerned, we did discuss this point in the manuscript in the end of section 4.3 (p 17) since we are aware that this aspect is of importance.

Thank you; we do agree that our database is not large enough to really assess impacts under future climates such as RCP2.6 and RCP8.5. In the revised manuscript, only RCP trends will be used in order to assess the sensitivity of our database to wilting point, P decrease, T increase, P decrease + T increase.

No conclusion : As mentioned in our reply to referee#1, a proper conclusion will be added at the end of the manuscript

Detailed comments

We agree with all other detailed comments / changes suggested but:

- P3, L5, References for empirical-based model: most of the empirical-based models use the same light and temperature dependences as those formulated by Guenther et al. (1995); currently, the most two widely used models are MEGAN (Guenther et al., 2006; 2012) and BEIS (Pier and Waldruff, 1991; Pierce et al., 1998). Emission estimates made using both models can often vary widely due to the differences in the land cover, emission factors and canopy models used. However models have not been found to outperform each other consistently when coupled to a air chemistry model and compared with measurements of some relevant atmospheric chemical species (e.g. ozone, aerosol, formaldehyde).

- P4, L3-8: indeed, the manuscript was structured in order to highlight the main scientific objectives of this study which were (i) to evaluate the in-situ impacts of ND and AD on Q. pub, in order (ii) to develop a parameterization to represent ER variability using 'simple' integrated environmental parameters, and eventually (iii) to assess ER sensitivity to P decrease and/or T increase in the future.

- P5, L22: Only a small fraction (<5%) of the data was missing ; of course, the bias between Pcum curves at both sites was assessed and considered to 'extrapolated' the missing values at the O3HP site. As the precipitations were cumulated over 7, 14, 21 days, the bias was negligible (around 1%).

- P6, L18: AF is dry leaf mass per area conversion factor, called LMA in a previous line. AF should hence be just changed by LMA.

We will check that all the abbreviations used are homogeneously cited.

- P8, L1, How Krustal-Wallis test detects seasonal variation? Kruskal-Wallis tests allowed to check for different median values among months. ANOVA tests could not be used since data did not follow the requirements of parametrical tests.
- P9, L31: we clarified earlier (fig a.) how the rain exclusion of 30% was achieved along the seasons.

- P10, L2-5: Although, in the absolute, the gap between Pcum(AD) and Pcum(ND) curves increases toward the end of 2012, from July 2012 the relative difference between both curves was maintained around 30 % (fig. 1c: 510 and 760 mm respectively, thus a 33 % reduction) .

- P11, L2, 'P=3.9' was a mistake.

- P11, L5-9: because the measurements were made right during the beginning of the isoprene onset period, some of the sampled branches started to emit isoprene significantly, while others emitted only at a very low level; this led to a large variability that could not be significantly related to the relative position of the branches in the AN or ND plots.

- P11, L13-14: these points were answered earlier in the General comments, section 'Results'

- P11, L18: indeed, as suggested by Referee #1 (Detailed comments, Figure caption) the regression lines and $R^2$ will be added in Fig. 3a and given in the text.

- P12, L6-7: our text was indeed not precise enough; we meant that under AD 'the contribution of the two lowest frequencies (-14 and -21 d.), was, RELATIVELY to the contribution of the two highest frequencies (instantaneous and -7d.), higher in April (48%) and June (700 and 40 % in 2012 and 2013 respectively) than during the summer (22, 8.5 and 16 % in July, Aug. and Sept. respectively)'.

- P15, L21-24: the 'MEGAN adjustment' to our local data being also raised by Referee#3, a detailed and common explanation is given in our reply to Referee#3 (last paragraph of 'main concerns').

- P16, L12: although our revised manuscript will focus onto the sensitivity of isoprene to soil moisture rather than on 'large scale predictions', we do think it is important to

remind, at least as a perspective in the new 'conclusion' section, that the reduction of the uncertainty on local/global isoprene emission assessments also require a better description of soil moisture model.

- Section 4.3: all result descriptions will be removed in order to tighten and focus this section onto discussion points only.

- P17: indeed, as mentioned in our answer to general comments, the revised manuscript will present sensitivity tests of ER to T, Pcum and SW, rather than future projections; because there is no other in situ long term study of drought effect on isoprene emissions, direct comparisons are not possible; however we will give more details on the findings obtained during a seasonal study carried out on isoprene emission from Q. pub. saplings by Genard-Zielinski et al. (2014) – already cited in our manuscript. We will then extend our discussion and compare our results to other in situ drought study in the Mediterranean area, but not for isoprene. For instance, Lavoir et al. (2009) reported lower monoterpene emission rates from Q. ilex in the rain exclusion plot from June to August, during the 2nd and 3rd year or rain exclusion. Since Q. ilex does not possess specific leaf reservoirs for monoterpene storage, emissions of Q. ilex respond to light and temperature as isoprene emissions. Their emission is hence de novo and tightly related to their synthesis. Monoterpene emissions of this species are thereby comparable to isoprene emissions.

- Figure1: we will add the x axis legend also on the top of the (a) graph in order to make the reading easier

- Figure 2: Indeed, differences between ND and AD using Mann-Whitney tests should be denoted with asterisks at the top of each figure. Differences among months were tested using Kruskal Wallis tests (W) followed by followed by Newman-Kruels post-hoc test with a<b<c<d. These corrections will be made.

Note that the Gw unit should be "mmol cm-2 s-1" and not "mol cm-2 s-1".

- Figure 3: unfortunately log scale was not suitable for this figure since many calculated values was set to zero by MEGAN; here after we present the log/log figures with the remaining non zero values (Fig. II)

- Figure 5: indeed, the '%T' and '%W' abbreviations are missing, as well as 'inst', '7d', '14d' and '21d'.

Referee #3

We do thank Referee #3 for her/his careful reading and useful comments and suggestions.

In order to make the reading smoother and the understanding clearer, the revised version of the manuscript will be proofread by a native English speaker before submitted a revised version. In addition, a special attention will be paid to better link the development of the different discussions.

Main concerns:

- Bibliographical weakness:

Although the references cited are not all 'narrow in focus and scope' and that a large number of references given do already focus on the effects of drought onto photosynthesis/cellular processes/isoprene emissions (Chaves et al., 2002; Lichtenthaler et al., 1997; Funk et al., 2004, Simon et al., 2005; Tani et al., 2011; Guenther et al., 2013; Wiberley et al., 2005; Llusià et al., 2008, 2009; Owen et al., 1998; Loreto & Schnitzler, 2010; Pegoraro et al., 2004) we will also cite the works of Sharkey and Loreto (1993) and Brili et al (2007).

Note that we already referred twice to the Peñuelas and Staudt (2010) paper (in the Introduction p2 and in the Discussion p16), which is a review, and thus, inherently, cannot be considered as narrow.

Although our manuscript do not aim at reviewing all the isoprene modelling studies

based on MEGAN, the excellent work of Müller et al. (2008) will be additionally considered and their assessments of the sensitivity of isoprene emissions to soil moisture will be presented; note however that Müller et al. (2008) reported that, when MEGAN model is evaluated against the 2 forest sites of Harvard (US) and Tapajos (Amazonia), it does 'fail to reproduce the observed seasonal variation at the tropical rainforest site' (wet and dry seasons).

The assessment of the wilting point teta(w) in the Sindelarova et al., 2014 study was based in the Pegoraro et al. (2004) work that we do already refer to in our discussion: as mentioned in our manuscript, Sindelarova et al. (2014) remind that under this threshold value, the isoprene emission is set to zero.

As we did observe isoprene emissions when soil moisture values were as low as 0.007 m3 m-3, we considered that testing the sensitivity of ER to teta(w) using so low values was pointless and not relevent regarding the soil type present at the O3HP; we did conclude that the Pegoraro approach was not valid for drought adapted tree species.

Although we did not mention it, we did assess the wilting point teta(w) at the 03HP to be 0.15 m3 m-3, a value very close to the 0.138 m3 m-3 value (Chen et Dudhia, 2001) we used; since our original manuscript was aiming at producing large scale projections, we made the choice to test MEGAN tuned with this teta(w) litterature data. Anyway, this slight difference of wilting point value does not change, in the end, the performance of MEGAN which cannot represent isoprene emissions for SW conditions much smaller thant the teta(w) value (MOST of the calculated ER remained set to zero).

Only when the SW effect was turned off in MEGAN (set to 1 for every calculation), were we able to test the effect of using on site isoprene emission factors Is rather than the Simpson et al. (1999) one (53 $\mu$gC gDW-1 h-1); we present and discuss later the results of such a comparison (see Fig III and corresponding text).

- Measurements:

1. We will make it clearer in the revised manuscript but :

- we did not aim at extrapolating – using MEGAN or another emission model – isoprene fluxes from the whole canopy; our focus was to study, for the first time in situ and over such a time scale, the ND and AD drought impacts on (only) sunlit branches of Q. pub; however some measurements were made under the canopy (see later on).

- thus, all the primary production, stomatal conductance and isoprene emission rates presented in this study were – only - measured for the leaves sampled at the top of the canopy ; the effect of the canopy structure is not addressed in this paper.

- the shading effect was investigated, only for 2 trees in the ND plot, and only in June 2012; we observed that, although the shading effect can be strong (PAR lowered by a factor of 6 to 18 ), measured isoprene emission factors (Is) were not significantly different (P>0.05) at the top and below the canopy (77+/-3 and 59+/-12 $\mu$g gDM-1 h-1 respectively); these points are detailed in another paper (Genard-Zielinski et al., 2015).

2. As noticed by Referee #2 our manuscript is lacking of clear details on how the roof was operated in order to simulate an AD. More information is given in our reply to Referee#2. Figure I shows that, during this study, the 30% precipitation reduction was reached quickly after the operation of the roof operation started, in May 2012.

When this study was conducted we were well aware that the future climatic changes will affect not only the intensity of the precipitations but the period over which these precipitations will occur. Since our study started just after the roof operation was available; we did focus, for this 1st year of measurements, only on the exclusion intensity aspect rather than its 'timing'.

3. We don't see any scientific basis for which Referee#3 could believe that we do relate the reality of our work. So, yes, a tremendous effort was made at the O3HP to conduct our field work and YES, we did sample Q. pubescens over one week per every month, as detailed in the Table below. Moreover we did not state that we sampled

isoprene 'one week per month from June 2012 till June 2013'; we did precise that 'measurements were performed at least during one week, once a month, from June 2012 to June 2013, except from November 2012 to March 2013 when Q. pubescent is fully senescent with leaves remaining on the tree' (P7, L26-28).

June 2012 : 1 - 16;

July 2012: 15 - 20;

August 2012:19 - 24;

September 2012:18 - 24;

October 2012: 22 - 26;

April 2013 : 22 - 27; June 2013: 16 - 22;

All isoprene measurements presented here were sampled on cartridges except during April 2013 where only on-line PTRMS was used (and no cartridges). An intercomparison between Cartridge+GCMS and PTRMS was carried on parallel on another emitter present on the site (Acer monspessulanum); no significant difference was observed between both techniques (see Genard-Zielinski, 2014 for details).

As mentioned earlier, this seasonal study was not planned to focus on vertical profile of BVOC through the canopy.

4. The COOPERATE data:

All the data (PAR, ambient T, Precipitation, SW, ST) used to train the ANN were obtained from the COOPERATE dataset; we used daily averaged values. This information will be added in the revised manuscript.

As explained earlier to referee#2, only a very few number (< 5 %) of precipitation data needed for assessing P integrated over 7, 14 or 21 days, was missing. As we spent quite a lot of time on the site, we are well aware that precipitations can, locally, be

highly variable in time and space; this is why we used the relative differences observed between the 2012 and 2013 Pcum curves in both sites (Forcalquier and O3HP) in order to 'fill the gap' of the missing data. Due to the fact that the P was integrated over 7, 14, 21 days, we assumed that this bias remains negligible compared to reducing the number of data in our ANN study due to missing P values.

Statistics

1. We are aware on how the 'G91' and 'G13' parameterizations lead to the integrated and much more complex emission model MEGAN; we know that MEGAN does include some 'historical' dependence of environmental conditions as we – too briefly – mentioned and discussed P15, L19-20. Thus, of course, we did consider the PAR and T over the previous 24 and 240 hours before our measurements in our calculated isoprene emissions (as we did consider and applied the leaf age dependence gamma(age), from eq(2) page 3189, Guenther et al., 2006)). We agree that these points should be more clearly stated. Moreover, we should more clearly remind that MEGAN does consider some 'historical' effect.

As we did not consider the canopy effect in our study, the canopy structure dependence as considered in MEGAN through the gamma(CE) parameter (from eq(2) page 3189, Guenther et al., 2006) was not considered (i.e. set to 1).

As we measured emission rates using a chamber but not canopy fluxes, we did not have to consider any canopy loss (we set gamma=1, from eq(1), page 3183, Guenther et al., 2006).

In the revised manuscript, and as also suggested by Referre#2, we shall more clearly detail these important aspects to help the reader to better understand how we tuned MEGAN for assessing our emission rates. We shall also emphasize our knowledge that MEGAN does account for the 'history' of the plant, better than we did on P15, L19-20.

As also requested by Referee#2 the CL CT parameterizations and a brief description of MEGAN will, thus, be presented in a new appendix.

2. We agree that our dataset is neither large enough nor robust enough for capturing future emission under RCP scenarios. As mentioned earlier and as suggested by Referee#2, the use of G14 will be limited in the revised version, to some sensitivity tests to temperature and/or drought changes.

3. Indeed ANN is often referred to as a 'grey' box, since such a statistical approach does not aim at providing some new mechanistic understanding of the studied process. On the other hand, when a mechanistic approach is failing to do so, ANN can provide some fruitful directions to where the 'mechanistic approaches' shall be looking at. In our case, some knowledge was gained by demonstrating, in particular, that the dependence of isoprene emissions under ND and AD (at least for Q. pub.):

- should be considered on a frequency lower than 10 days (as in MEGAN) and at least up to 3 weeks, and not only for L or T parameters

- are related to ST (soil temperature) as well, highlighting – probably - the possible impact of the soil microorganism functioning on isoprene emissions

- cannot be assessed, for drought adapted plant, by a 'threshold' type approach such as in MEGAN.

Eventually, note that in MEGAN, the isoprene dependency to PPFD over the previous 24 and 240h was deduced by a 'best statistical fit' (P3190, L28-32 Guenther et al., 2006), without providing any precise mechanistic insight on how and why it is working; the useful of MEGAN is, nevertheless, not challenged.

Concerning the use of site-specific data, we already answered and discussed earlier to this point; the figures III below present the effects of using our on-site isoprene emission factors Is rather than the value of 53 given by Simpson et al. (1999), WHEN on only WHEN no SW effect was considered in MEGAN. We observe that :

- in the control plot (ND), less variability was represented by MEGAN when on-site Is were used (fig c) compared to Is=53 (fig a) (39% compared to 46% respectively) and that the overall underestimation was reduced (from 47 to 26 % respectively)

- in the stress plot (AD): only 40% of the variability was represented by MEGAN, whatever the Is used; an over-estimation of 18% was obtained when on-site Is were used (fig d) compared to an underestimation of more than a factor 2 when Is was set to 53 (fig b). Although these results do not represent the full architecture of MEGAN (SW effect was not activated) they could be presented and discussed in the revised manuscript.

ORCHIDEE

As explained P9, L8, ORCHIDEE was run to assessed SW and ST for the RCP exercises.

As explained P8, l26 to P9 L6, ORCHIDEE used ISI-MIP met data for RCP projections.

Our projections were not made specifically for the O3HP site, but for a Mediterranean site representative of the O3HP site conditions. Therefore, precipitations –or other parameters – were not downscaled to the O3HP site; they are representative of the whole cell which corresponds to the one where the O3HP site is located.

Recommendations

Site-specific data: this point is discussed several times above.

We remind once more that the wilting point was assessed for the O3HP site; when used, MEGAN does not agree better with our observations.

As explained earlier, physiological parameters such as sap flow, transpiration, . . . were not measured during this work; our study aimed at relating our observations to 'more simple', more 'integrative' environmental variable, 'easily' accessible than complex physiological data; we hypothesized that, anyway, sap flow, transpiration, . . . are in the end, more or less indirectly, driven by L, T, SW, ST considered over a large range

of time.

Detailed comments

None

New references cited in our responses

Sharkey, T. D. and Loreto, F.: Water stress, temperature, and light effects on the capacity for isoprene emission and photosynthesis of kudzu leaves, Oecologia, 95, 328–333, 1993.

Brilli, F., Barta, C., Fortunati, A., Lerdau, M., Loreto, F., and Centritto, M.: Response of isoprene emission and carbon metabolism to drought in white poplar (Populus alba) saplings, New Phytol., 175, 244–254, 2007.

A.-V. Lavoir1, M. Staudt1, J. P. Schnitzler2, D. Landais1, F. Massol3, A. Rocheteau4, R. Rodriguez1, I. Zimmer2, and S. Rambal1, Drought reduced monoterpene emissions from the evergreen Mediterranean oak Quercus ilex: results from a through fall displacement experiment, Biogeosciences, 6, 1167–1180, 2009

Please also note the supplement to this comment:
https://www.biogeosciences-discuss.net/bg-2017-17/bg-2017-17-AC1-supplement.pdf

[Figure]

Fig. I: Averaged cumulated precipitation at the O$_3$HP between 1967 and 2000 (blue line with upper and lower standard deviation), and during the driest years of the same period (red line with upper and lower standard deviation). Cumulated precipitation in 2012 in the rain exclusion plot is represented by the green line and in the exclusion plot by the purple line.

**Fig. 1.**

Figure II

[Figure]

[Figure]

**Fig. 2.**

Figure III

[Figure]

[Figure]

**Fig. 3.**

---

## Author Response (AR1)

**1. General responses to Referee reviews**

We thank the referees for their thorough review and their useful suggestions for improving both, the presentation of our work, and the interpretation of our data.

We also thank the Editor for extending twice the deadline for submitting our responses.

In order to fulfill the major referees' requirements, the revised manuscript has been deeply changed and re-written; in particular:

- A better focus on our (new) scientific objectives is now presented in the revised introduction;
- The sequencing and the link between the different sections has been improved;
- A better description of how we used the MEGAN model is now given in a new specific section (section 2.5); in particular, we explain how we did run MEGAN2.1 using our in situ data (wilting point, seasonal emission factors, …);
- We present the sensitivity of MEGAN2.1 performance over a large range of wilting point values;
- We now give some suggestions on how MEGAN model could be improved in order to better account for drought stress of drought adapter isoprene emitters;
- As we agreed that our dataset was neither large nor robust enough for capturing future emission under RCP scenarios, the '2100 projections' are no more considered; instead, we present the sensitivity of *Q. pub.* isoprene emissions to expected future climate changes for 6 different cases: moderate and severe T changes, moderate and severe P changes, moderate and severe T+P changes;
- Further information has been added, in particular, in the 'Materials and Methods' concerning which version of MEGAN was used, the ANN limitation and uncertainty, how the exclusion system was operated during our experiment, how ORCHIDEE model was used and what for, and how the COOPERATE dataset was used;
- A conclusion has been added

Consequently:

- The sections 3.3 and 3.4 ('Results') and 4.2 and 4.3 ('Discussion') were completed re-written;
- A new title is now proposed to better fit with the scope of the revised manuscript ("Seasonal variations of *Q. pubescens* isoprene emissions from an *in natura* forest under drought stress and sensitivity to future climate change in the Mediterranean area");
- The order and title of some sections of the 'Materials and Methods' has been changed, and a new section 2.5 is now dedicated to the version of MEGAN we used;
- Except for Fig. 1 & 2, all other figures are new, and 2 tables have been added;

Eventually, in order to make the reading smoother and the understanding clearer the manuscript has been proofread (see certificate).

**2. Responses to Referee #1**

**2a. General comments**
None

**2b. Detailed comments**
We do thank Referee #1 for his/her careful review and comments.

**Abstract**

Line 1: 'physiology' has been changed by 'gas exchange' (and all along the document as well).

Lin 20-23: the lowest ER values in October and April, 6 and <2 µg gDW$^{-1}$ h$^{-1}$ respectively, have been added.

Lin 20: no, the lowest Gw values were observed between July and September (<20 mole$_{H2O}$ m$^{-2}$ s$^{-1}$) not in April (figure 2b).

Line 22: we do not understand this point; emission rates ER (measured values) are different from emission factors Is (=ER normalized to temperature and PAR).

**Materials and Methods**

Pg 5 Lin 10: sampling volumes varied between 0.45 and 0.9 L depending on the season and the hour of the day, thus, on the expected emission intensity. This precision is now given in the revised section 2.2.

Pg 8 Lin 7: a new web site link is now given in the revised section 2.7.

**Results**

Pg 9 lin 31: no, PAR peaks in June 13 (899.3 µmole m-2 s-1).

Pg 10 Lin 20: further information is now available in the revised sections 3.2 and 4.1 (Electric resistivity tomography measurements have shown the heterogeneity of the karstic substrate organized as soil pockets developed between limestone rocks. Water and nutrient pools and dynamics probably differ greatly between the shallow upper soil layers and the soil pockets developed between limestone rocks. However, the soil trenches carried out in the site have shown that a calcareous slab often developed at a depth of 10-20 cm and that the roots of the oaks were rather distributed in this superficial humiferous horizon, and that only a few large roots cross this slab.)

Pg 10, Lin 26: Senescence of leaves: although senescence had just begun during this sampling period, but we did check that the enclosed branches were not senescent during our measurements. This information is now given in the revised section 2.2.

**Discussion**

Point 1: As already mentioned, the water availability from deep roots is now better discussed in the revised discussion, section 4.1.

Point 2: we do consider moderate to more severe drought effects; this is why we mentioned the overall range of drought intensity.

Point 3: we suggest that the frequency over which the different environmental parameters (not only light) should be larger than the one considered so far, and how these frequencies are changing over the year (see new section 4.2 and conclusion).

Pg 13, lin 15: although this section has been completely re-written we made sure to put Q. Pub in italics in the document.

Pg 15, lin 12: this section has been completely re-written.

Pg 16, lin 9-16: this section has been completely re-written.

**Figure caption**

Same colors of the legends in all graphs: we used blue and red colors for ND and AD respectively in all graphs; therefore, we are not quite sure to understand what referee#1 wants.

Pg 23, lin 3: July 2013 was changed to June 2013.

Pg 23, lin 9 & 18: 'June 2013' was added.

Pg 23, lin 18: Since Fig. 3 has been changed a new figure caption is now given.

Fig 1: in Fig. 1 (and other figures as well) the colors do differ from ND (blue) and AD (red) treatment. However, PAR and T values being the same for both plots, this color code was deliberately not used for PAR and T.
Legend axis was reissued as suggested and a vertical line between 2012 and 2013 was added.

Fig 2: we did not add 'isoprene' as suggested to keep the title short; ER and Is meanings are given in the figure caption and, anyway, all the paper is about isoprene (and not another VOC) emissions.

Fig 3: Former figure 3 was changed by a new one.
Fig 4: Former figure 4 was removed.
Fig 5: Former figure 5 was removed.

**3. Responses to Referee #2**

We do thank Referee #2 for her/his careful reading and useful comments and suggestions.

**3a. General comments**

- Use of literature values instead of site-observed data: G14 was tuned using O3HP data, not literature data; this point is now more clearly explained in the new section 2.5.

- No discussion of potential uncertainties from G14 and ANN in general: details are now given in the revised section 2.6 and discussed in the new section 4.3. (Among the other available statistical methods, ANNs present the advantage of being the most parsimonious (e.g. giving the smallest error for a same number of descriptors; see for instance Dreyfus et al., 2002). Moreover, ANN approach, as the other non-linear regression methods, is not, or not very, sensitive to regressors' co-linearity (Bishop, 1995; Dreyfus et al., 2002). One of the ANNs limitations is that they can be used only for interpolation, not extrapolation exercises. For this reason, our future RCP projections we made using only xi values that did fit into the range of variation of xi obtained during the training phase; in total 21% of data were thus rejected. ANN optimization during the training phase was based on the reduction of the root mean square error (RMSE) between calculated and measured values. Our final optimized RMSE (validation data) was 8.5 µg gDW$^{-1}$ h$^{-1}$ for our ER values ranging between 0.06 and 113 µg gDW$^{-1}$ h$^{-1}$, and represents 35% of the mean (22.7 µg gDW$^{-1}$ h$^{-1}$)).

- No comparison with process based model: as mentioned in the first paragraph, p3 of the original document, process based model require a complex set of data to be ran; such a dataset is not available for this study. This is the reason why an 'empirical' model (MEGAN) was solely tested.

- No conclusion and no clear key messages: a new conclusion has been added, and key messages are now given.

- Abstract:

Too many abbreviations: we cannot use less than the 4 remaining abbreviations (ER, ND, AD, G14) if we want to keep the abstract as concise as possible.

Sentences were added at the beginning of the abstract in order to better explain 'why we need this study'.

- Method:

No description of MEGAN: a complete description of the MEGAN2.1 model and how it was tuned in our study is now given in a new specific section 2.5.

Better description on how ORCHIDEE data were used and how G14 was tuned: the purpose of running ORCHIDEE and how the ISI-MIP and ORHCIDEE derived data were used in our G14 algorithm is now detailed in sections 2.6, 2.7, 2.8 and 3.4.

A more detailed description on how AD roof was operated is now given in section 2.1.

- Results:

Use of on-site data to apply MEGAN rather than 'literature' data: detailed information is now given in the new section 2.5.

- Discussion:

Long text on future ER although no consideration on C source + training period for G14 is too short: C source was indeed not directly considered since the hypothesis we made in applying a neuronal

approach to statistically analyze our data was to consider 'simple' integrative environmental parameters; C source, nor vegetation adaptation, were thus not considered, but *in fine* we hypothesized that they are, more or less indirectly, driven by PAR, T, SW and ST fluctuations over a range of different frequencies.

As vegetation adaptation is concerned, we did discuss this point at the end of the original section 4.3 (p 17) since we are aware that this aspect is of importance. This point is also mentioned in our revised and new section 4.3.

Thank you: we do agree that our database was not large enough to really assess impacts under future climates such as RCP2.6 and RCP8.5. In the revised manuscript, the future climatic data are now used only to test the sensibility of isoprene emissions to T, P and T+P changes.

No conclusion: As mentioned earlier a proper conclusion is now given at the end of the manuscript.

**3b. Detailed comments**

- 'gaz' has been changed to 'gas'.
- P3, L5, References for empirical-based model: Ashworth et al. (2013) is actually a review where both types of models (empirical and process based) are described.

- P3, L31 and P3, L32: this part of the introduction has been changed, but we made sure that to explain an abbreviation when used for the first time.

- P4, L3-8: thank you; the scientific questions considered in our work are now better explained at the end of the introduction, and in the "results" and "discussion".

- P4, L26: this sentence has been rewritten at the beginning of the revised section 2.2.

- P4, L27: 'except from' has been changed as requested in the revised section 2.2.

- P5, L22: only a small fraction (<5%) of the data was missing ; of course, the bias between Pcum curves at both sites was assessed and considered to 'extrapolated' the missing values at the O3HP site. As the precipitations were cumulated over 7, 14, 21 days, the bias was negligible (around 1%) and no adjustment was made. This information has been added in the revised section 2.7.

- P6, L18: AF is dry leaf mass per area conversion factor, called LMA in the previous line. AF was hence changed by LMA in the revised section 2.3 (former section 2.4).

 All the abbreviations cited were homogenized, in the text and in the figures.

- P6, L23: Cout and Cin order has been changed in the revised section 2.3.

- P8, L1: How Krustal-Wallis test detects seasonal variation? Kruskal-Wallis tests allowed to check for different median values among months.  ANOVA tests could not be used since data did not follow the requirements of parametrical tests. This information has been added in the revised section 2.4 (former section 2.5).

- P9, L31: we clarified in the revised section 2.1 how the rain exclusion of 30% was achieved along the seasons.

- P10, L2-5:  although, in the absolute, the gap between Pcum(AD) and Pcum(ND)  curves increases toward the end of 2012, from July 2012 the relative difference between both curves was maintained at a ≈ 30 % (fig. 1c: 510 and 760 mm respectively, thus a 33 % reduction) .

- P10, L11/P5 L17-19: we clarified in the revised section 3.1, why, although PAR measured in the COOPERATE database was made on one point above the canopy, the PAR actually received by the enclosed branch in the chamber could be different from one branch to another, especially in September 2012,during the autumnal sun setting.

- P10, L22-23: this line has been removed

- P11, L2: 'P=3.9' was a mistake and was removed.

- P11, L5-9: because the measurements were made right during the beginning of the isoprene onset period, some of the sampled branches started to emit isoprene significantly, while others emitted only at a very low level; this led to a large variability that could not be significantly related to the relative position of the branches in the AN or ND plots. Further precision is now given in the revised section 3.2.

- P11, L13-14: these points have been already mentioned in our general responses and are now detailed in the new section2.5 dedicated to MEGAN 2.1 description.

- P11, L18: specific regression lines/$R^2$ are now given when needed in the document and/or in the Fig or Tables.

- P12, L6-7: thank you, our text was indeed not precise enough; we meant that under AD 'the contribution of the two lowest frequencies (-14 and -21 d.), was, RELATIVELY to the contribution of the two highest frequencies (instantaneous and -7d.), higher in April (48%) and June (700 and 40 % in 2012 and 2013 respectively) than during the summer (22, 8.5 and 16 % in July, Aug. and Sept. respectively)'. The text has been changed accordingly in the revised section 3.3.

- P12, L9: the contribution to each frequency has been converted to ratio and is now presented in the revised fig. 6 (former Fig. 4).

Note that the discussion concerning the relative contribution of each environmental parameter is no longer made: indeed, SW appeared as the 'major' driving regressors tested in G14 … since this was the ONLY regressor that differed between both sample sets (ND and AD).

- P12, L30: this section has been completely re-written, but we made sure to use correct number of RCPs in the revised text.

- P12, L31; P13, L17: these parts of the text have been completely re-written.

- P14, L5: thank you, 'µgC gDM-1 h-1' is now the only unit used for emission rates throughout in the revised text.

- P14 L28-31 and P15, L21-24: as explained earlier, and in our General Responses, the 'MEGAN adjustment' to our local data is now explained in the new section 2.5.

- P16 L2-3: 'degree day' unit was changed to 'Dd' and explained when it appeared for the first time in the revised section 4.2.

- P16, L12: soil water description by land models is no more addressed in the revised manuscript.

- Section 4.3: as mentioned in our General Responses, this section has been completely re-written. As suggested, sensitivity tests are now presented.

Concerning the comparison with other literature data, in the new conclusion we are now insisting on the fact that there is actually no in situ isoprene data comparable to our O3HP database; however, in the new section 4.3 we are now giving more details on the findings obtained during

a seasonal study carried out on isoprene emission from *Q. pub.* saplings by Genard-Zielinski et al. (2015). In order to extend our discussion on species emitting other BVOC than isoprene and compare our results to other in situ drought study in the Mediterranean area, we are now reporting in the revised section 4.1 the Lavoir et al. (2009) study.

- Figure1: an x axis legend has been added also on the top of the (a) graph in order to make the reading easier.

- Figure 2: letter 'c' and 'd' meaning is now added in the revised figure caption of figure2.

Indeed, differences between ND and AD using Mann-Whitney tests should be denoted with asterisks at the top of each figure. Differences among months were tested using Kruskal Wallis tests (W) followed by followed by Newman-Kruels post-hoc test with a<b<c<d. These corrections have been made in the revised Figure caption.

Note that the Gw unit should be 'mmol $cm^{-2}$ $s^{-1}$' as indicated in the revised document and not 'mol $cm^{-2}$ $s^{-1}$' as previously written.

- Figure 3: former Fig. 3 was removed and replaced by a new one.

However, on this figure and others, log scale was not always suitable since, as explained in the discussion, many ER were set to zero by MEGAN2.1.

- Figure 5: Former Fig. 5 was removed and replaced by a new one.

**4.  Responses to Referee #3**

We do thank Referee #3 for her/his thorough reading and useful suggestions. We tried, in particular:

- to better explain how we tuned MEGAN2.1 model in order to assess its ability in reproducing the effect of drought on Q. Pub ER ;
- to study the sensitivity of MEGAN2.1 to the wilting point value;
- to suggest some improvement in the formulation of the drought activity factor used in MEGAN2.1.

**4a. Main concerns:**

- Bibliographical weakness:

  Although many of the references cited in our manuscript were not all narrow in focus and scope, and that a large number of them did focus on the effects of drought onto photosynthesis/cellular processes/isoprene emissions (Chaves et al., 2002; Lichtenthaler et al., 1997; Funk et al., 2004, Simon et al., 2005; Tani et al., 2011; Guenther et al., 2013; Wiberley et al., 2005; Llusià et al., 2008, 2009; Owen et al., 1998; Loreto and Schnitzler, 2010; Pegoraro et al., 2004), we have added the works of Sharkey and Loreto (1993), and Brili et al. (2007).

  In the original paper, we did refer twice to the Peñuelas and Staudt (2010) paper (in the Introduction p2 and in the Discussion p16 of the former version), which is a review, and inherently, cannot be considered as narrow in scope.

  Our manuscript does not aim at reviewing all the isoprene modelling studies based on MEGAN; however the excellent work of Müller et al. (2008) is now cited in the revised manuscript, and their assessments of the sensitivity of isoprene emissions to soil moisture is now mentioned in the revised section 3.3.

  As now clearly explained (in the revised introduction and discussion), the assessment of the wilting point $\theta w$ in the Sindelarova et al. (2014) study was based on the Pegoraro et al. (2004) work.

- Measurements:

  1. We have now made it clear in the revised manuscript (new section 2.5) that:

  - we did not aim at extrapolating – using MEGAN or any another emission model – isoprene fluxes from the whole canopy; our focus was to test MEGAN in assessing the impact of ND and AD on isoprene emission rates as observed from the sunlit branches of Q. pub; the effect of the canopy structure is thus not addressed in this paper (even if some measurements were made under the canopy during this study, see next paragraph).

  - the shading effect was investigated, only for 2 trees in the ND plot, and only in June 2012; we observed that, although the shading effect can be strong (PAR lowered by a factor of 6 to 18 ), measured isoprene emission factors (Is) were not significantly different (P>0.05) at the top and below the canopy (77$\pm$3 and 59$\pm$12 µg gDM$^{-1}$ h$^{-1}$ respectively); these points have been detailed in another paper (Genard-Zielinski et al., 2015).

  2. We give in the revised section 2.2 a full description on how the roof was operated in order to simulate an AD.

  When this study was conducted we were well aware that the future climatic changes will affect not only the intensity of the precipitations but its seasonality too (see our discussion on that

point). Since our study started just after the roof operation was available, we did focus, for this 1st year of measurements, only on the exclusion intensity aspect rather than its 'timing'.

3. We don't see any scientific basis for which Referee#3 could state that we do relate the reality of our field work. Yes, a tremendous effort was made at the O3HP to conduct our field work and yes, we did sample Q. pub. over one week per month, as detailed in the table below.

| Month | Sampling dates |
|---|---|
| June 2012 | 1 - 16 |
| July 2012 | 15 - 20 |
| August 2012 | 19 - 24 |
| September 2012 | 18 - 24 |
| October 2012 | 22 - 26 |
| April 2013 | 22 - 27 |
| June 2013 | 16 - 22 |

Note that we did not state that we sampled isoprene 'one week per month from June 2012 till June 2013'; we did precise that 'measurements were performed at least during one week, once a month, from June 2012 to June 2013, except from November 2012 to March 2013 when Q. pubescent is fully senescent with leaves remaining on the tree' (P7, L26-28).

All isoprene measurements presented here were sampled on cartridges except during April 2013 where only on-line PTRMS was used (and no cartridges). An intercomparison between Cartridge+GCMS and PTRMS was carried on parallel on another emitter present on the site (Acer monspessulanum); no significant difference was observed between both techniques. This information is now given in the revised section 2.3.

4. The COOPERATE data:

As now mentioned in the revised section 3.3, ANN training and validation were carried out using data measured at the O3HP (sampled ER, COOPERATE environmental data); we now explain in the revised manuscript that daily averaged values were used when regressors were cumulated over 7-21 days.

As mentioned in our response to referee#2, only a very few number (< 5 %) of precipitation data needed for assessing P integrated over 7, 14 or 21 days, was missing. As we spent quite a lot of time on the field site, we are well aware that precipitations can, locally, be highly variable in time and space; this is why we used the relative differences observed between the 2012 and 2013 Pcum curves in both sites (Forcalquier and O3HP) in order to 'fill the gap' of the missing data. Due to the fact that the P was integrated over 7, 14, 21 days, this bias remains negligible. These detailed are now given in the revised section 2.7.

Statistics

1. Being aware that MEGAN does include some 'historical' dependence of environmental conditions (as mentioned P15, L19-20 of the original manuscript) we did consider the PAR and T over the previous 24 and 240 hours before our measurements when we calculated isoprene emissions. This information is now clarified in the new section 2.5.

The other different points raised by Referee#3 concerning how we used MEGAN (canopy structure, canopy loss) are also detailed in the new section 2.5.

2. As mentioned earlier, no more 2100 projections are presented in the revised manuscript.

3. Indeed ANNs are often referred to as a 'grey' box, since such a statistical approach does not aim at providing new mechanistic understandings of the studied process. On the other hand, when a mechanistic approach is failing to do so, ANNs can provide some fruitful information. Note in particular that in MEGAN the isoprene dependency to PPFD over the previous 24 and 240h was deduced by a 'best statistical fit' (P3190, L28-32 Guenther et al., 2006), without providing any precise mechanistic insight on how and why it is working; the useful of MEGAN is, nevertheless, not challenged.

In our case, by using G14 we showed that, for a drought adapted emitter, isoprene emission shall be much more sensitive to T than to P under future climatic changes; MEGAN2.1 was unable to do so, since not 'adapted' to account for drought effect of a drought adapted isoprene emitter.

ORCHIDEE
A better description of what data were obtained by ORCHIDEE and how these data were used in G14 is now given in the revised sections 2.7, and 3.4.
Our projections were not made specifically for the O3HP site, but for a Mediterranean site representative of the O3HP site conditions. Therefore, precipitations –or other parameters – were not downscaled to the O3HP site.

**4b. Recommendations**

We followed the final recommendations made by the referee#3:

- As it is now explained in the new section 2.5, site-specific data were used for MEGAN2.1 assessment.

- Site data were also used for G14 development (as explained in the revised section 3.3)

- Site data (COOPERATE) were also used in G14 to assess the present case in the sensitivity tests (see revised section 3.4)

- The sensitivity of MEGAN2.1 to the wilting point is now presented in the revised sections 3.3 and 4.2.

However, as explained in the original manuscript, and as it is now mentioned twice in the revised manuscript, physiological parameters such as sap flow, transpiration, … were not measured during this work since it was not the aim of this work. Rather than using such complex data, we favored the use of more 'integrative' environmental variable, 'easily' accessible than complex physiological data; we hypothesized that, *in fine*, sap flow, transpiration, … are more or less indirectly driven by L, T, SW, ST, especially if they are considered over a large of frequency range as in G14.

**5. New references cited in our responses**

Brili, F., Barta, C., Fortunati, A., Lerdau, M., Loreto, F., and Centritto M.: Response of isoprene emission and carbon metabolism to drought in white poplar (*Populus alba*) saplings, New Phytol., 175(2), 244-254, 2007.

Lavoir, A.V., Staudt, M., Schnitzler, J.P., Landais, D., Massol, F., Rocheteau, A., Rodriguez, R., Zimmer, I., and Rambal, S.: Drought reduced monoterpene emissions from the evergreen Mediterranean oak *Quercus ilex* : results from a throughfall displacement experiment, Biogeosciences, 6, 1167-1180, 2009.

Sharkey, T. D., and Loreto, F.: Water-stress, temperature, and light effects on the capacity for isoprene emission and photosynthesis of kudzu leaves, Oecologia, 95, 328–333, 1993.

[Figure]

Proof-Reading-Service.com Ltd, Devonshire
Business Centre, Works Road, Letchworth Garden
City, Hertfordshire, SG6 1GJ, United Kingdom
Office phone: +44(0)20 31 500 431
E-mail: enquiries@proof-reading-service.com
Internet: http://www.proof-reading-service.com
VAT registration number: 911 4788 21
Company registration number: 8391405

29 March 2018

To whom it may concern,

**RE: Proof-Reading-Service.com Editorial Certification**

This is to confirm that the document described below has been submitted to Proof-Reading-Service.com for editing and proofreading.

We certify that the editor has corrected the document, ensured consistency of the spelling, grammar and punctuation, and checked the format of the sub-headings, bibliographical references, tables, figures etc. The editor has further checked that the document is formatted according to the style guide supplied by the author. If no style guide was supplied, the editor has corrected the references in accordance with the style that appeared to be prevalent in the document and imposed internal consistency, at least, on the format.

It is up to the author to accept, reject or respond to any changes, corrections, suggestions and recommendations made by the editor. This often involves the need to add or complete bibliographical references and respond to any comments made by the editor, in particular regarding clarification of the text or the need for further information or explanation.

We are one of the largest proofreading and editing services worldwide for research documents, covering all academic areas including Engineering, Medicine, Physical and Biological Sciences, Social Sciences, Economics, Law, Management and the Humanities. All our editors are native English speakers and educated at least to Master's degree level (many hold a PhD) with extensive university and scientific editorial experience.

**Document title:** **Simulating precipitation decline under a Mediterranean deciduous Oak forest: effects on isoprene seasonal emissions and predictions under climatic scenarios**

**Author(s):** **Anne-Cyrielle Genard-Zielinski et al.**

**Format:** **British English**

**Style guide:** **Biogeosciences at https://www.biogeosciences.net/index.html**

[revised manuscript text omitted]

---

## Referee Report (RR1)

Comments on
"Seasonal variations of *Q. pubescens* isoprene emissions from an *in natura* forest under drought stress and sensitivity to future climate change in the Mediterranean area"
by Anne-Cyrielle Genard-Zielinski et al.

This work is a comprehensive study started from site experiment/measurement, evaluated model performance, developed a new model and evaluated future impact on isoprene emissions over Mediterranean region and is a contribution to scientific research in this field. This paper investigated the impact of drought on isoprene emissions of Q. pubescens. Although the response of ER to drought is not significant, emission factor increased considerably. Inadequacies of MEGAN2.1 are discovered and a new formulation of $\gamma_{sm}$ is suggested. ANN trained model G14 showed improved performance under certain conditions as well. Temperature and precipitation changes according to RCP scenarios are found to contribute to ER in the future.

At this stage of revision, the paper is well written, clear and well structured. There are a few comments or questions I would love to have the authors clarify in the paper:

1) In chapter 2.6, 3.3 and appendix 1, G14 algorithm is introduced and shows a significant performance improvement in simulating ER with environmental inputs. Could you explain a bit more on how overtraining issue is tested and avoided in this practice?
Why is July an outlier in this algorithm? Is there an explanation for that, or does it suggest some potential issues with certain measurements over this period?

2) When applying RCP projections as inputs for G14 algorithm, 21% of the data were rejected. What is the rejection criteria, especially variables related to temperature? Do you only use data that are in the range of G14 training dataset for all of the inputs, or majority of them? Please clarify.

3) Figure 6 illustrates the relative contribution of different regressor frequencies, however, regressors at certain frequencies are not picked as predictors in G14, for instance, T-7, ST-1, P-1. How is this relative contribution calculated, when they are not all included in the algorithm?

**Minor comments:**
P1L21, (+30%, AD), suggest to change to (around 30%, AD) or (AD, 32%).
P2L18, "Brili et al., 2007Loreto and Fineschi, 2015" missing comma and space.
P8L2, $\varepsilon_{iso,Qp}$ is not introduced here or before.
P8L7, PPFD is not introduced.
P10L5, N=7 neurons? Or layers of neurons? Please clarify.

---

## Author Response (AR2)

**Referee #1**

We do thank Referee #1 for his/her careful review and comments. Here are our point-to-point reply.

**General comments**

1)

**Overtraining phenomena**: overtraining was checked by comparing $RMSE_{training}$ and $RMSE_{validation}$ for each value of $N$ (1 to 7) tested; overtraining was observed when $RMSE_{validation}$ became superior to $RMSE_{training}$ after $N$=3 (figure below); this precision is now given section 2.6, P10L11-12.

[Figure]

Additionally, in order to make sure that obtained EQM did not correspond to local error minimum, a large number of iterations (n=300) were selected for every ANN run.

**Outlying July data**:

- sampling/analysis were carried out in July similarly to other months: we could not, thus, detected any measurements issues;

- observed July ER were already strongly uncoupled with the $C_L \times C_T$ both in the ND and AD (R²=0,06 and 0,01 respectively, see section 3.2) showing that other parameters than $L0$ and $T0$ were strongly controlling the emission variations. Among the (many!) different ANN structures tested during this study, we noticed that when July data were (only slightly) better represented, the ANN performances on the other data were generally lower, leading to an overall higher RMSE. Since July measurements corresponded probably to the transition when the ND sub-set of data started to be significantly different from the AD sub-set of data (physiological adaptation), this short episode may have been statistically under-represented in our dataset to be correctly described by an ANN approach. As a compromise, we decided to 'forget' the poor agreement noticed for July in order to maintain a good agreement on all the other data. More measurements on a similar period would improve our database and the overall ANN ability to describe such 'outlier' data.

Additional precision is now given in section 3.3, P15L20-22.

2) Rejection criteria: for all the input regressors, we actually accepted all the RCP data that were in the range of the training dataset ± 20%. This precision is now given at the end of section 2.6, P10L21.

3) Figure 6: only frequencies were considered to produce this figure, not the type of predictors. Thus, the class "0", "-7", "-14", "-21" included the contribution of "$L$-1, $T$-1, SW-1, L0, T0, $T_M$-$T_m$", "SW7, ST7, $P$7", "$T$14, SW14, ST14, $P$14" and "$T$21, SW21, $P$21" respectively. The figure 6 caption was changed in order to better explain how the figure was made.

**Minor comments**

P1L21: "(+30%, AD) has been changed to "(AD, 32%), P1L22.

P2L18: comma and space has been added, P2L19.

P8L2: $\varepsilon_{iso,Qp}$ is now introduced in the section 2.4 when it first appeared, P8L4.

P8L17: PPFD is now introduced P8L20.

P10L5: only one layer was tested; this precision is now given, section 2.6, P10L8.

[revised manuscript text omitted]

Measured ER ($\mu gC\ g_{DM}^{-1}\ h^{-1}$)

**Figure 6**

[Figure]

**Figure 7**

[Figure]

**Figure 8**

[Figure]

**Figure 9**

[Figure]

**Table 1**

| $x$ | ND | | AD | |
|---|---|---|---|---|
| | ER/ER$_{MEGAN}$=f($x$) | R² value | ER/ER$_{MEGAN}$=f($x$) | R² value |
| SW | $0.653e^{10.5x}$ | 0.13 | $0.192e^{51.1x}$ | 0.66 |
| SW-7 | $0.715e^{1.30x}$ | 0.13 | $0.239e^{6.30x}$ | 0.55 |
| SW-14 | $0.763e^{0.57x}$ | 0.11 | $0.279e^{2.74x}$ | 0.48 |
| SW-21 | $0.523e^{0.46x}$ | 0.14 | $0.365e^{1.47x}$ | 0.38 |

**Table 2**

| | ΔSW (m³ m⁻³) | Δ*Pcum* (mm) | ΔST (° C) | Δ*T* | ΔSW/SW | Δ*Pcum*/*Pcum* (%) | ΔST/ST | Δ*T*/*T* |
|---|---|---|---|---|---|---|---|---|
| RCP2.6 | + 0.004 | + 30 | + 1.4 | + 1.4 | + 0.5 | +5 | + 8.4 | + 9.1 |
| RCP8.5 | - 0.007 | + 30 | + 5.3 | + 5.3 | - 5.0 | -24 | + 32 | + 34 |